# AUCSeg: AUC-oriented Pixel-level Long-tail Semantic Segmentation

**Boyu Han**[1,2]     **Qianqian Xu**[1,3*]     **Zhiyong Yang**[2]     **Shilong Bao**[2]
**Peisong Wen**[1,2]     **Yangbangyan Jiang**[2]     **Qingming Huang**[2,1,4*]

[1] Key Lab. of Intelligent Information Processing, Institute of Computing Technology, CAS
[2] School of Computer Science and Tech., University of Chinese Academy of Sciences
[3] Peng Cheng Laboratory
[4] Key Laboratory of Big Data Mining and Knowledge Management, CAS
{hanboyu23z,xuqianqian,wenpeisong20z}@ict.ac.cn,
{yangzhiyong21,baoshilong,jiangyangbangyan,qmhuang}@ucas.ac.cn

## Abstract

The *Area Under the ROC Curve (AUC)* is a well-known metric for evaluating instance-level long-tail learning problems. In the past two decades, many AUC optimization methods have been proposed to improve model performance under long-tail distributions. In this paper, we explore AUC optimization methods in the context of pixel-level long-tail semantic segmentation, a much more complicated scenario. This task introduces two major challenges for AUC optimization techniques. On one hand, AUC optimization in a pixel-level task involves complex coupling across loss terms, with structured inner-image and pairwise inter-image dependencies, complicating theoretical analysis. On the other hand, we find that mini-batch estimation of AUC loss in this case requires a larger batch size, resulting in an unaffordable space complexity. To address these issues, we develop a pixel-level AUC loss function and conduct a dependency-graph-based theoretical analysis of the algorithm's generalization ability. Additionally, we design a *Tail-Classes Memory Bank (T-Memory Bank)* to manage the significant memory demand. Finally, comprehensive experiments across various benchmarks confirm the effectiveness of our proposed AUCSeg method. The code is available at https://github.com/boyuh/AUCSeg.

## 1 Introduction

Semantic segmentation aims to categorize each pixel within an image into a specific class, which is a fundamental task in image processing and computer vision [33, 51, 75]. Over the past decades, substantial efforts [58, 23, 43, 96] have advanced the field of semantic segmentation. The mainstream paradigm is to develop innovative network architectures that encode more discriminative features for dense pixel-level classifications. Typical backbones include CNN-based [58, 14, 80] and newly emerging Transformer-based methods [108, 83, 65, 15, 32], which have achieved the state-of-the-art (SOTA) performance. Beyond this direction, researchers [24, 66, 48, 41] have recently realized the ***Pixel-level Long-tail issue* in Semantic Segmentation (PLSS)**, as shown at the top of Figure 1. Similar to the flaws of traditional long-tail problems, the major classes will dominate the model learning process, causing the model to overlook the segmentation of minority classes in an image. Several remedies have been proposed to alleviate this [49, 61, 10, 56, 104, 92, 78]. For example, [77] introduces a category-wise variation technique inversely proportional to distribution to achieve balanced segmentation; [66] introduces a sequence-based generative adversarial network for imbalanced medical image segmentation, and [41] develops a re-weighting scheme for semi-supervised segmentation.

---

*Corresponding authors.

38th Conference on Neural Information Processing Systems (NeurIPS 2024).

Currently, mainstream studies fall into two camps. One is to develop carefully designed backbones for long-tail distributions but leave the effect of loss functions unconsidered. The other is to conduct empirical studies on the loss functions without exploring their theoretical impact on the generalization performance. A question then arises naturally:

> *Can we find a theoretically grounded loss function for PLSS on top of SOTA backbones?*

This paper provides an affirmative answer from the AUC perspective and proposes a novel framework called *AUC-oriented Pixel-level Long-tail Semantic Segmentation (AUC-Seg)*. Specifically, AUC indicates the likelihood that a positive sample scores higher than a negative one, which has been proven to be **insensitive** to data distribution [86, 101]. Applying AUC to **instance-level long-tail** classifications has shown promising progress in the machine learning community [86, 100, 88, 68]. Motivated by its success, this paper starts an early trial to study AUC optimization for PLSS. The primary concern is to study its effectiveness for PLSS from a theoretical perspective. The key challenge is that the standard techniques for generalization analysis [62, 8, 21] require the loss function to be expressed as a sum of independent terms. Unfortunately, the proposed loss function does not satisfy this assumption due to the dual effect of structured inner-image dependency and pairwise inter-image dependency. This complicated structure poses a big challenge to understanding its generalization behavior. To address this, we decompose the loss function into inner-image and inter-image terms. On top of this reformulation, we deploy the dependency

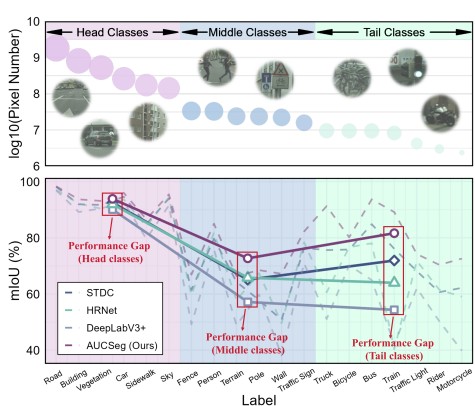

Figure 1: Statistic information of pixel number for each class in the Cityscapes training set and the performance of previous methods (DeepLabV3+, HRNet and STDC) compared to our method (AUC-Seg). Our method aims to improve overall performance, particularly for tail classes. The dashed lines represent mIoU values for each class, while the solid lines represent the average mIoU for the head, middle, and tail classes.

graph [103] to decouple the interdependency. Finally, we reach a bound of $\widetilde{\mathcal{O}}\left(\tau\sqrt{\log\left(\tau Nk\right)/N}\right)$, where $\tau$ behaves like an indicator for imbalance degree, and $k$ denotes the number of pixels in each image. This suggests optimizing AUC loss could ensure a promising performance under PLSS.

Back to the practical AUC learning process, we realize that the stochastic gradient optimization (SGD) for structured pixel-level tasks imposes a greater computational burden compared to instance-level long-tail problems. Specifically, the SGD algorithm of AUC requires **at least one sample from each class in each mini-batch** [99, 85]. In light of this, the primary choice is to adopt the so-called stratified sampling on all images [70, 60, 86] for mini-batch generation (See Equation (7)). Unfortunately, as shown in Figure 3(a) and (b), this is hard to implement under PLSS because **pixel-level labels are densely coupled in each image**. Meanwhile, as shown in Proposition 1, we also argue that directly using random sampling to include all classes would require an extremely large batch size. This leads to unaffordable GPU memory demands for optimization, as described in the experiments in Appendix G.6.

To alleviate this, a novel *Tail-class Memory Bank (T-Memory Bank)* is carefully designed. The main idea is to identify those missing pixel-level classes in each randomly generated mini-batch and then complete these absences using stored historical class information from the T-Memory Bank. This enables efficient optimization of AUCSeg with a light memory usage, enhancing the scalability of our proposed method, as shown in Figure 1. Finally, comprehensive empirical studies consistently speak to the efficacy of our proposed AUCSeg.

Our main contributions are summarized as follows:

- This paper starts the first attempt to explore the potential of AUC optimization in pixel-level long-tail problems.

- We theoretically demonstrate the generalization performance of AUCSeg in semantic segmentation. To our knowledge, this area remains underexplored in the machine-learning community.

- We introduce a Tail-class Memory Bank to reduce the optimization burden for pixel-level AUC learning.

## 2 Related Work

### 2.1 Semantic Segmentation

Semantic segmentation is a subtask of computer vision, which has seen significant development since the inception of FCN [58]. The most common framework for semantic segmentation networks is the encoder-decoder. For the encoder, researchers typically use general models such as ResNet [37] and ResNeXt [84]. As the segmentation tasks become more challenging, some specialized networks have emerged, such as HRNet [75], ICNet [105], and multimodal networks [76, 42]. For the decoder, a series of studies focus on strengthening edge features [23, 107], capturing global context [43, 31, 45], and enhancing the receptive field [96, 67, 12, 13]. Recently, the transformer has shown immense potential, surpassing previous methods. A series of methods related to Vision Transformer [108, 83, 65, 15, 74] are proposed. SegNeXt [32], which is the current *state-of-the-art (SOTA)* method, possesses the same powerful feature extraction capabilities as the Vision Transformer and the same low computational requirements as CNN. Apart from improving the network, some research [48, 56, 24, 11, 10, 66, 39] is directed toward addressing the issue of class imbalance in semantic segmentation. However, the effectiveness of these methods is not significant. In this paper, we aim to improve the performance of long-tailed semantic segmentation from an AUC optimization perspective.

### 2.2 AUC Optimization

The development of AUC Optimization can be divided into two periods: the machine learning era and the deep learning era. As a pioneering study, [20] ushers in the era of AUC in machine learning. It studies the necessity of AUC research, which points out that AUC maximization and error rate minimization are inconsistent. After that, AUC gains significant attention in linear fields such as Logistic Regression [38] and SVM [46, 47]. Then researchers begin to explore the online [106, 28] and stochastic [94, 63] optimization extensions of the AUC maximization problem. Research from the perspectives of generalization analysis [2, 73, 17] and consistency analysis [1, 29] provides theoretical support for AUC optimization algorithms. [57] is the first to extend AUC optimization to deep neural networks, ushering in the era of AUC in deep learning. Meanwhile, a series of AUC variants [87, 86, 68, 88, 69, 90] emerge, gradually enriching AUC optimization algorithms. Furthermore, in practice, AUC optimization demonstrates its effectiveness in various class-imbalanced tasks, such as recommendation systems [5, 6, 4, 7], disease prediction [79, 30], domain adaptation [89], and adversarial training [40, 91].

Despite significant progress, existing studies of AUC optimization mainly pay attention to the instance-level imbalanced classification tasks. This paper starts an early trial to introduce AUC optimization to semantic segmentation. However, due to the high complexity of pixel-level multi-class AUC optimization, such a goal cannot be attained by simply using the current techniques in the AUC community.

## 3 Preliminaries

In this section, we briefly introduce the semantic segmentation task and the AUC optimization problem.

### 3.1 Semantic Segmentation Training Framework

Let $\mathcal{D} = \{(\mathbf{X}^i, \mathbf{Y}^i)_{i=1}^n | \mathbf{X}^i \in \mathbb{R}^{H \times W \times 3}, \mathbf{Y}^i \in \mathbb{R}^{H \times W \times K}\}$ be the training dataset, where $H$ and $W$ represent the height and width of the images, and $K$ denotes the total number of classes. Let $f_\theta$ be a semantic segmentation model ($\theta$ is the model parameters), which commonly follows an encoder-decoder backbone [75, 105, 108, 83, 32]. Let $\hat{\mathbf{Y}}^i = f_\theta(\mathbf{X}^i) \in \mathbb{R}^{H \times W \times K}$ be the dense

pixel-level prediction, *i.e.* ,

$$\hat{\mathbf{Y}}^i = f_\theta(\mathbf{X}^i) = f_\theta^d(f_\theta^e(\mathbf{X}^i)), \tag{1}$$

where the encoder $f_\theta^e$ extracts features from the image $\mathbf{X}^i$, and then the decoder $f_\theta^d$ predicts each pixel based on extracted features and outputs a dense segmentation map with the same size as $\mathbf{Y}^i$.

Furthermore, let $\mathbf{Y}_{u,v}^i$ and $\hat{\mathbf{Y}}_{u,v}^i$ represent the ground truth and prediction of the $(u, v)$-th pixel of the $i$-th image, respectively. To train the model $f_\theta$, most current studies [14, 80, 83, 65] usually adopt the *cross-entropy (CE) loss*:

$$\ell_{ce} := \frac{1}{n} \sum_{i=1}^{n} \sum_{u=0}^{H-1} \sum_{v=0}^{W-1} \left[ -\sum_{c=1}^{K} \mathbf{Y}_{u,v}^{ic} \log(\hat{\mathbf{Y}}_{u,v}^{ic}) \right], \tag{2}$$

where $\mathbf{Y}_{u,v}^{ic}$ and $\hat{\mathbf{Y}}_{u,v}^{ic}$ are the one-hot encoding of ground truth and the prediction of pixel $(\mathbf{X}_{u,v}^i, \mathbf{Y}_{u,v}^i)$ in class $c$, respectively.

### 3.2 AUC Optimization

*Area under the Receiver Operating Characteristic Curve (AUC)* is a well-known ranking performance metric for **binary classification** task, which measures the probability that a positive instance has a higher score than a negative one [35]:

$$AUC(f_\theta) = \mathbb{P}\left( f_\theta(\mathbf{X}^+) > f_\theta(\mathbf{X}^-) | y^+ = 1, y^- = 0 \right), \tag{3}$$

where $(\mathbf{X}^+, y^+)$ and $(\mathbf{X}^-, y^-)$ represent positive and negative samples, respectively. When $AUC \to 1$, it indicates that the classifier can perfectly separate positive and negative samples.

According to [87, 86, 88], given finite datasets, maximizing $AUC(h_\theta)$ is usually realized by maximizing its unbiased empirical estimation:

$$\hat{AUC}(h_\theta) = 1 - \frac{1}{n^+ n^-} \sum_{i=1}^{n^+} \sum_{j=1}^{n^-} \ell(h_\theta(\mathbf{X}^+) - h_\theta(\mathbf{X}^-)), \tag{4}$$

where $\ell$ is a differentiable surrogate loss [86] measuring the ranking error between two samples, $n^+$ and $n^-$ denote the number of positive and negative samples, respectively.

Moreover, we can directly optimize the following problem for AUC maximization:

$$\min_\theta \frac{1}{n^+ n^-} \sum_{i=1}^{n^+} \sum_{j=1}^{n^-} \ell(h_\theta(\mathbf{X}^+) - h_\theta(\mathbf{X}^-)). \tag{5}$$

Note that, AUC has achieved significant progress in long-tailed classification [100, 88, 68]. Due to the limitations of space, we refer interested readers to the literature [86, 99] for more introductions to AUC. However, most existing studies merely focus on the **instance-level or image-level** problems. Inspired by its distribution-insensitive property [26], this paper starts an early trial to introduce AUC to PLSS.

## 4 AUC-Oriented Semantic Segmentation

In this section, we introduce our proposed AUCSeg method for semantic segmentation. A brief overview is provided in Figure 2. AUCSeg is a generic optimization method that can be directly applied to any SOTA backbone for semantic segmentation. Specifically, AUCSeg includes two crucial components: **(1) AUC optimization** where a theoretically grounded loss function is explored for PLSS and **(2) Tail-class Memory Bank**, an effective augmentation scheme to ensure efficient optimization of the proposed AUC loss. In what follows, we will go into more detail about them. For clarity, we include a table of symbol definitions in Appendix A.

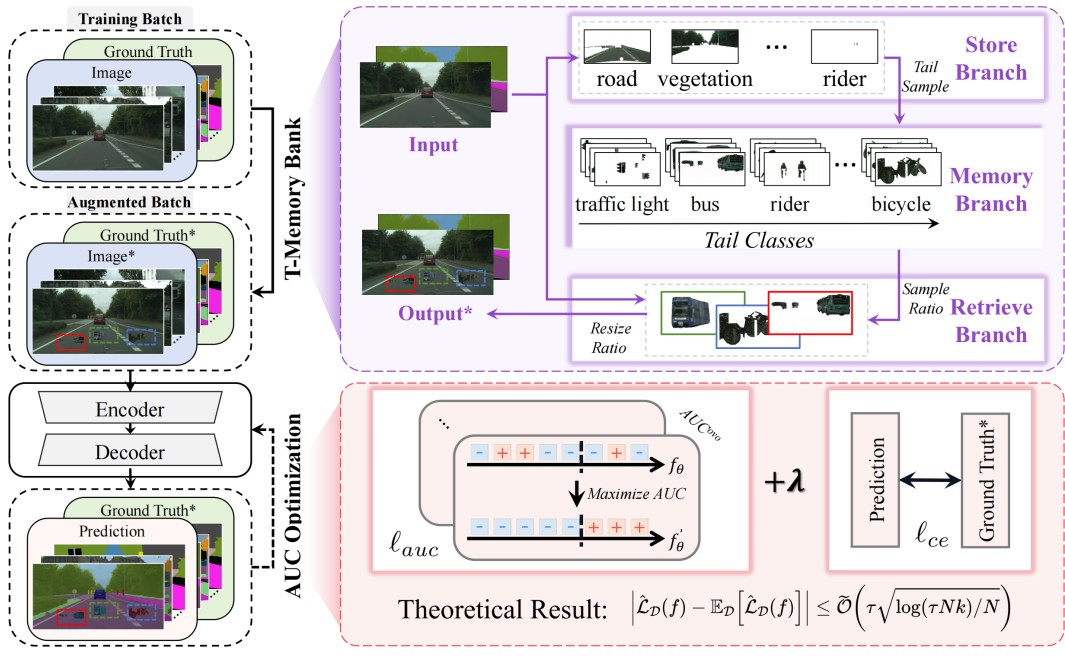

Figure 2: An overview of AUCSeg.

## 4.1 Pixel-level AUC Optimization

Semantic segmentation is a multi-class classification task. Therefore, to apply AUC, we follow a popular multi-class AUC manner, *i.e.* , the One vs. One (ovo) strategy [64, 34, 86], which is an average of binary AUC score introduced in Section 3.2. Specifically, on top of the notation of Section 3.1, we further denote $\mathcal{D}^p = \{(\mathbf{X}_{u,v}^i, \mathbf{Y}_{u,v}^i) | i \in [1, n], u \in [0, H-1], v \in [0, W-1]\}$ as the set of all pixels; the $j$-th element ($j \in [1, n \times (H-1) \times (W-1)]$) in $\mathcal{D}^p$ is abbreviated as $(\mathbf{X}_j^p, \mathbf{Y}_j^p)$ for convenience. Given the model prediction $f_\theta = (f_\theta^{(1)}, \ldots, f_\theta^{(K)})$, $\forall c \in [K], f_\theta^{(c)} \in [0, 1]$, where $f_\theta^{(c)}$ serves as a continuous score function supporting class $c$, $AUC_{seg}^{ovo}$ calculates the average of binary AUC scores for every class pair:

$$AUC_{seg}^{ovo} = \frac{1}{K(K-1)} \sum_{c=1}^{K} \sum_{c \neq c'} AUC_{cc'}(f_\theta), \qquad (6)$$

$$AUC_{cc'}(f_\theta) = \mathbb{P}(f_\theta^{(c)}(\mathbf{X}_m^p) > f_\theta^{(c)}(\mathbf{X}_n^p) | \mathbf{Y}_m^p = c, \mathbf{Y}_n^p = c').$$

To this end, as introduced in Section 3.2, the goal is to minimize the following unbiased empirical risk:

$$\ell_{auc} := \sum_{c=1}^{K} \sum_{c' \neq c} \sum_{\mathbf{X}_m^p \in \mathcal{N}_c} \sum_{\mathbf{X}_n^p \in \mathcal{N}_{c'}} \frac{1}{|\mathcal{N}_c||\mathcal{N}_{c'}|} \ell_{sq}^{c,c',m,n}, \qquad (7)$$

where we adopt the widely used square loss $\ell_{sq}(x) = (1-x)^2$ as the surrogate loss [29]; $\ell_{sq}^{c,c',m,n} :=$ $\ell_{sq}(f_\theta^{(c)}(\mathbf{X}_m^p) - f_\theta^{(c)}(\mathbf{X}_n^p))$; $\mathcal{N}_c = \{\mathbf{X}_k^p | \mathbf{Y}_k^p = c\}$ represents the set of pixels with label $c$ in the set $\mathcal{D}^p$, and $|\mathcal{N}_c|$ denotes the size of the set.

## 4.2 Generalization Bound

In this section, we explore the theoretical guarantees of the AUC loss function in semantic segmentation tasks and demonstrate that AUCSeg can generalize well to unseen data.

A key challenge is that standard techniques for generalization analysis [62, 8, 21] require the loss function to be expressed as a sum of independent terms. Unfortunately, the proposed loss function does

not satisfy this assumption because there are **two layers of interdependency among the loss terms**. On one hand, semantic segmentation can be considered a structured prediction problem [16], where couplings between output substructures within a given image create the first layer of interdependency. On the other hand, the AUC loss creates a pairwise coupling between positive and negative pixels, so any pixel pairs sharing the same positive/negative instance are interdependent, resulting in the second layer of interdependency.

We present our main result in the following theorem and the proof is deferred to Appendix B.

**Theorem 1** (Generalization Bound for AUCSeg). *Let $\mathbb{E}_{\mathcal{D}}\left[\hat{\mathcal{L}}_{\mathcal{D}}\left(f\right)\right]$ be the population risk of $\hat{\mathcal{L}}_{\mathcal{D}}\left(f\right)$. Assume $\mathcal{F} \subseteq \{f : \mathcal{X} \to \mathbb{R}^{H \times W \times K}\}$, where $H$ and $W$ represent the height and width of the image, and $K$ represents the number of categories, $\hat{\mathcal{L}}^{(i)}$ is the risk over $i$-th sample, and is $\mu$-Lipschitz with respect to the $l_\infty$ norm, (i.e. $\|\hat{\mathcal{L}}(x) - \hat{\mathcal{L}}(y)\|_\infty \leq \mu \cdot \|x - y\|_\infty$). There exists three constants $A > 0$, $B > 0$ and $C > 0$, the following generalization bound holds with probability at least $1 - \delta$ over a random draw of i.i.d training data (at the image-level):*

$$\left|\hat{\mathcal{L}}_{\mathcal{D}}\left(f\right) - \mathbb{E}_{\mathcal{D}}\left[\hat{\mathcal{L}}_{\mathcal{D}}\left(f\right)\right]\right| \leq \frac{8}{N} + \frac{\eta_{inner} + \eta_{inter}}{\sqrt{N}}\sqrt{A\log\left(2B\mu\tau Nk + C\right)}$$
$$+ 3\left(\sqrt{\frac{1}{2N}} + K\sqrt{1 - \frac{1}{N}}\right)\sqrt{\log\left(\frac{4K(K-1)}{\delta}\right)},$$

*where*

$$\eta_{inner} = \frac{48\mu\tau\ln N}{N}, \quad \eta_{inter} = 2\sqrt{2}\tau, \quad \tau = \left(\max_{c \in [K]} \frac{n_{\max}^{(c)}}{n_{mean}^{(c)}}\right)^2,$$

*$n_{\max}^{(c)} = \max_{\mathbf{X}} n(\mathbf{X}^{(c)})$, $n_{mean}^{(c)} = \sum_{i=1}^{N} n(\mathbf{X}_i^{(c)})$, $N = |\mathcal{D}|$, $k = H \times W$ and $\mathbf{X}^{(c)}$ represents the pixel of class $c$ in image $\mathbf{X}$.*

**Remark 1.** We achieve a bound of $\widetilde{\mathcal{O}}\left(\tau\sqrt{\log\left(\tau Nk\right)/N}\right)$, indicating reliable generalization with a large training set. Here, $\tau$ represents the degree of pixel-level imbalance. More interestingly, even though we have $k$ classifiers for every single image, the generalization bound only has an algorithm dependent on $k$, suggesting that pixel-level prediction doesn't hurt generalization too much.

### 4.3 Tail-class Memory Bank

**Motivation.** Although we have examined the effectiveness of AUC for PLSS from the theoretical point of view, there is **a practical challenge** when conducting AUC optimization for semantic segmentation, as discussed in Section 1. Specifically, the stochastic AUC optimization, as defined in Equation (7), requires **at least one sample from each class** in a mini-batch. In **instance-level AUC optimization**, recent studies [87, 86] often use a stratified sampling technique that generates batches consistent with the original class distribution, as shown in Figure 3(a). Such a strategy will work well when each image belongs to a unique category (say, 'Banana', 'Apple', or 'Lemon') in traditional classifications. Yet it cannot apply to pixel-level cases because each sample involves multiple and coupled labels, making it hard to split them for stratified sampling, as illustrated in Figure 3(b). Meanwhile, we also provide a bound (Proposition 1) to show that simply adopting random sampling will suffer from an overlarge batch size $B$, making an unaffordable GPU memory burden.

**Proposition 1.** *Consider a dataset $\mathcal{D}$ that includes images with $K$ different pixel categories. Let $p_i$ represent the probability of observing a pixel with label $i$ in a given image. Randomly select $B$ images from $\mathcal{D}$ as training data, where*

$$B = \Omega\left(\frac{\log(\delta/K)}{\log(1 - \min_i p_i)}\right).$$

*Then with probability at least $1 - \delta$, for any $c \in [K]$, there exists $\mathbf{X}$ in the training data that contains pixels of label $c$.*

**Remark 2.** The proof is deferred to Appendix C. Proposition 1 suggests that the value of $B$ is inversely proportional to $\min_i p_i$. Note that $p_i$ will be smaller as the long-tail degree becomes more

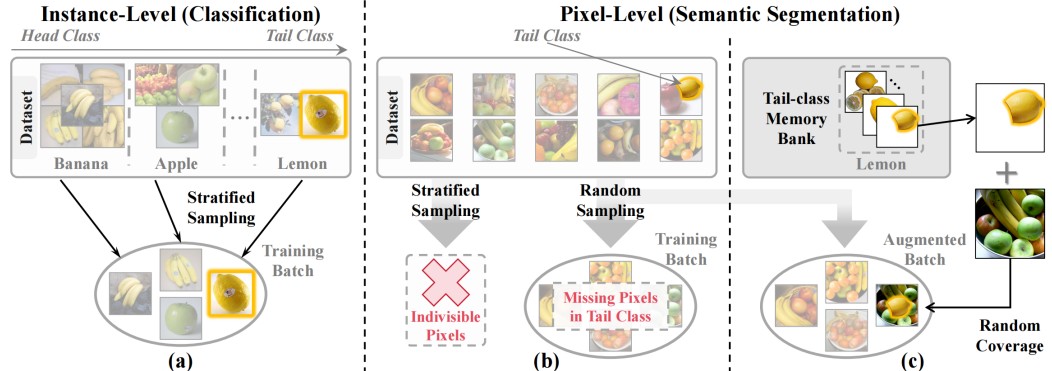

Figure 3: Instance-level and pixel-level task sampling.

severe, leading to a larger $B$. For example, in terms of the Cityscapes dataset with $K = 19$ classes, assuming $\delta = 0.01$ and $\min_i p_i = 1\%$, $B$ should be at least 759 to guarantee that each class of pixels appears at least once with a high probability. This results in a significant strain on GPU memory.

To address this, considering that the tail-class samples generally have less opportunity to be included in a mini-batch and are often more crucial for final performance, we thus develop a novel Tail-class Memory Bank (T-Memory Bank) to efficiently optimize Equation (7) and manage GPU usage effectively. As depicted in Figure 3(c), the high-level ideas of the T-Memory Bank are as follows: 1) identify missing tail classes of all images involved in a mini-batch and 2) randomly replace some pixels in the image with missing classes based on stored historical class information in T-Memory Bank. In this sense, we can obtain an approximated batch-version of Equation (7), *i.e.* ,

$$\tilde{\ell}_{auc} := \sum_{c=1}^{K} \sum_{\substack{c' \neq c \\ n_c n_{c'} \neq 0}} \sum_{\substack{\mathbf{X}_m^p \in \mathcal{N}_c \cup \mathcal{T}_c \\ \mathbf{X}_n^p \in \mathcal{N}_{c'} \cup \mathcal{T}_{c'}}} \frac{1}{|\mathcal{N}_c||\mathcal{N}_{c'}|} \tilde{\ell}_{sq}^{c,c',m,n} \tag{8}$$

where $\mathcal{N}_c$ and $\mathcal{T}_c$ represent the set of pixels with label $c$ in the original image and those pixels stored in the T-Memory Bank, respectively; $\tilde{\ell}_{sq}^{c,c',m,n} := \ell_{sq}(f_\theta^{(c)}(\tilde{\mathbf{X}}_m^p) - f_\theta^{(c)}(\tilde{\mathbf{X}}_n^p))$; $\tilde{\mathbf{X}}^p$ represents the sample after replacing some pixels with tail classes pixels from the T-Memory Bank.

**Detailed Components.** As shown in Figure 2, T-Memory Bank comprises three main parts: **(1) Memory Branch** stores a set with $S_M$ (the Memory Size) images for each tail class. We define the set as $\mathcal{M} = \{\mathcal{M}_{c_1}, \ldots, \mathcal{M}_{c_{n_t}}\}$, where $\mathcal{C}_t = \{c_i\}_{i=1}^{n_t}$ denotes the labels of tail classes, and $n_t$ is the total number of selected tail classes; **(2) Retrieve Branch** selects pixels from the Memory Branch to supplement the missing tail classes and **(3) Store Branch** updates the Memory Branch whenever a new image arrives. Algorithm 1 summarizes a short version of AUCSeg equipped with T-Memory Bank. **Please refer to the detailed version in Appendix D.** Note that we introduce CE loss as a regularization term for our proposed AUCSeg, which is widely used in the AUC community [99] to pursue robust feature learning. Experiments demonstrate that the performance is insensitive to the regularization weight $\lambda$, as shown in Figure 5d.

At the start of training, the Memory Branch is empty. In this case, we only calculate the loss function $\ell$ for the classes present in the mini-batch, while the Retrieve Branch will not take any action. Meanwhile, the Store Branch will continuously append pixel data of tail classes to the Memory Branch. As the Memory Branch reaches its maximum capacity $S_M$, we adopt a random replacement strategy to update the Store Branch (Lines 5 to 6 in Algorithm 1 or Lines 6 to 11 in Algorithm 2).

As the training process progresses, if the Memory Branch is not empty, the Retrieve Branch kicks in to count the missing classes in each image of the mini-batch, denoted as $\mathcal{C}_{miss}$. It then calculates the number of classes needed to be added for optimization, $n_{sample} = \lceil |\mathcal{C}_{miss}| \times R_S \rceil$. Here, we introduce a tunable sample ratio $R_S$ to strike a trade-off between the original and missing tail-class semantic information. Finally, it uniformly retrieves the corresponding pixels of $n_{sample}$ missing classes from the Memory Branch, resizes them by the resize ratio $R_R$, and randomly selects positions to overwrite (Lines 7 to 8 in Algorithm 1 or Lines 13 to 18 in Algorithm 2).

---

**Algorithm 1:** AUCSeg Algorithm (Short Version)

---

**Input:** Training data $\mathcal{D}$, number of tail classes $n_t$, labels of tail classes $\mathcal{C}_t = \{c_i\}_{i=1}^{n_t}$, Memory Branch $\mathcal{M} = \{\mathcal{M}_{c_1}, \ldots \mathcal{M}_{c_{n_t}}\}$, memory size $S_M$, sample ratio $R_S$, resize ratio $R_R$, max iteration $T_{max}$, batch size $N_b$

**Output:** model parameters $\theta$

1 **for** $iter = 1$ *to* $T_{max}$ **do**
2    $\mathcal{D}_B = \{(\mathbf{X}_i, \mathbf{Y}_i)\}_{i=1}^{N_b} \leftarrow SampleBatch(\mathcal{D}, N_b)$;
3    $\mathcal{C}_{miss} \subseteq \mathcal{C}_t \leftarrow MissingTailClasses(\mathcal{D}_B)$;
4    $\overline{\mathcal{C}_{miss}} = \mathcal{C}_t - \mathcal{C}_{miss}$;
5    $\triangleright$**for** $\overline{c_m}$ *in* $\overline{\mathcal{C}_{miss}}$ **do**
6      Extract pixels of class $\overline{c_m}$ from $\mathcal{D}_B$ and save them to $\mathcal{M}_{\overline{c_m}} \leftarrow RandomReplace(S_M)$;
7    $\triangleright$**for** $i = 1$ *to* $\lceil |\mathcal{C}_{miss}| \times R_S \rceil$ **do**
8      Randomly choose $c_m$ from $\mathcal{C}_{miss}$ and sample pixels from $\mathcal{M}_{c_m}$ to paste into $\mathcal{D}_B \leftarrow SizeScale(S_R)$;
9    $\triangleright$ Calculate $\hat{\mathbf{Y}}_i = f_\theta(\mathbf{X}_i)$ and $\ell = \tilde{\ell}_{auc} + \lambda \ell_{ce}$;
10    Backpropagation updates $\theta$.

---

**Discussions.** We recognize that Memory Bank [82, 36] has achieved great success in deep learning. However, our T-Memory Bank behaves differently compared to earlier studies. **The key difference is that the Memory Bank and T-Memory Bank are designed for different tasks.** The goal of the previous Memory Bank is to facilitate the traditional classifications by storing instance-level or image-level features, while our T-Memory Bank specifically stores the original pixels for each object. This strategy is particularly beneficial for our PLSS task. Additionally, the T-Memory Bank enables AUCSeg without substantially increasing GPU workload as well as the number of samples per mini-batch by selectively replacing non-essential pixels. Our experiments, detailed in Appendix G.6, include a comparison of GPU overhead. For more discussion on the T-Memory Bank, please refer to Appendix E.

## 5 Experiments

In this section, we describe some details of the experiments and present our results. **Due to space limitations, please refer to Appendix F, Appendix G and Appendix H for an extended version.**

### 5.1 Experimental Setups

The experiment includes three benchmark datasets: Cityscapes [19], ADE20K [109], and COCO-Stuff 164K [9]. We use SegNeXt [32] as the backbone for our model and the *mean of Intersection over Union (mIoU)* as the evaluation metric. We compare our method with 13 **recent advancements and** 6 **long-tail approaches** in semantic segmentation. All the long-tail methods also use SegNeXt as the backbone. To ensure fairness, we re-implement the listed methods using their publicly shared code and test them on the same hardware. Detailed introductions are deferred to Appendix F.

### 5.2 Overall Performance

Table 1 shows the quantitative performance comparisons. We draw the following conclusions: First, most current algorithms perform poorly in long-tail scenarios. Specifically, performance drops sharply from head to tail classes. For instance, the performance gap for PointRend and OCRNet on the Cityscapes reaches up to $40\%$. Second, models using long-tail approaches generally achieve better results than those that do not. However, they still fail to produce satisfactory outcomes. One possible reason is that these long-tail approaches focus on reweighting, giving too much attention to the tail classes and leading to overfitting. Additionally, our proposed AUCSeg method surpasses all competitors in most metrics. This success is due to the appealing properties of AUC. Our method consistently outperforms the runner-up by $+1.21\%$, $+0.75\%$, and $+0.38\%$ in tail classes mIoU across the datasets. Overall mIoU also improves by $+1.05\%$, $+0.27\%$, and $+0.31\%$. In some Head/Middle

Table 1: Quantitative results on **Cityscapes**, **ADE20K** and **COCO-Stuff 164K** val set in terms of mIoU (%). The champion and the runner-up are highlighted in **bold** and underline.

| Method | ADE20K | | | | Cityscapes | | | | COCO-Stuff 164K | | | |
|---|---|---|---|---|---|---|---|---|---|---|---|---|
| | Overall | Head | Middle | Tail | Overall | Head | Middle | Tail | Overall | Head | Middle | Tail |
| DeepLabV3+ [14] | 31.95 | 75.88 | 51.96 | 26.01 | 66.53 | 90.11 | 57.16 | 54.36 | 29.11 | 51.11 | 32.93 | 24.82 |
| EncNet [102] | 32.12 | 75.34 | 51.60 | 26.32 | 71.34 | 91.62 | 60.76 | 63.03 | 27.31 | 49.89 | 30.41 | 23.09 |
| FastFCN [80] | 29.78 | 74.20 | 49.44 | 23.86 | 63.97 | 90.37 | 52.43 | 51.22 | 28.37 | 50.60 | 32.52 | 23.96 |
| EMANet [55] | 32.83 | 75.77 | 50.03 | 27.36 | 70.93 | 91.69 | 60.61 | 61.97 | 28.48 | 49.73 | 29.97 | 24.85 |
| DANet [27] | 33.83 | 74.62 | 51.01 | 28.52 | 65.77 | 89.66 | 55.30 | 54.26 | 26.83 | 49.60 | 31.14 | 22.29 |
| HRNet [71] | 31.83 | 75.35 | 49.98 | 26.19 | 73.40 | 91.98 | 65.79 | 64.00 | 28.65 | 48.00 | 30.74 | 25.16 |
| OCRNet [97] | 29.64 | 74.00 | 49.40 | 23.72 | 66.95 | 90.24 | 63.18 | 50.21 | 28.67 | 51.04 | 32.41 | 24.33 |
| DNLNet [93] | 33.24 | 75.90 | 51.16 | 27.69 | 70.68 | 91.98 | 59.90 | 61.66 | 30.23 | 50.71 | 33.05 | 26.41 |
| PointRend [50] | 17.77 | 67.18 | 37.60 | 11.46 | 60.67 | 89.79 | 53.92 | 41.49 | 11.17 | 21.17 | 13.64 | 9.04 |
| BiSeNetV2 [95] | 10.26 | 60.38 | 28.72 | 4.10 | 73.04 | 92.00 | 63.52 | 64.93 | 10.30 | 34.96 | 12.71 | 5.92 |
| ISANet [98] | 29.53 | 74.34 | 48.77 | 23.64 | 70.63 | 91.67 | 61.50 | 60.43 | 26.37 | 48.87 | 30.78 | 21.86 |
| STDC [25] | 30.17 | 73.36 | 48.02 | 24.58 | 76.30 | 92.58 | 65.09 | 71.94 | 29.83 | 51.74 | 33.40 | 25.61 |
| SegNeXt [32] | 47.45 | 80.54 | 60.35 | 43.28 | 82.41 | 94.08 | 72.46 | 80.92 | 42.42 | 57.05 | 41.71 | 40.33 |
| VS [49] | 24.72 | 75.30 | 48.02 | 17.86 | 55.40 | 92.16 | 52.52 | 26.36 | 24.27 | 47.80 | 30.38 | 19.19 |
| LA [61] | 31.16 | 77.07 | 53.43 | 24.77 | 62.75 | 92.98 | 64.79 | 35.09 | 28.56 | 49.67 | 33.16 | 24.21 |
| LDAM [10] | 33.11 | 74.06 | 51.26 | 27.65 | 65.95 | 92.72 | 69.27 | 40.17 | 42.39 | 56.85 | 41.59 | 40.34 |
| Focal Loss [56] | 47.68 | 80.54 | 59.04 | 43.73 | 82.44 | 93.90 | 72.79 | 80.89 | 41.98 | 56.87 | 41.51 | 39.79 |
| DisAlign [104] | 48.15 | 80.33 | 59.14 | 44.31 | 81.94 | 93.61 | 72.12 | 80.36 | 42.10 | 55.20 | 41.24 | 40.28 |
| BLV [77] | 46.76 | 79.96 | 58.96 | 42.67 | 81.81 | 93.84 | 71.83 | 80.05 | 42.17 | 56.83 | 41.52 | 40.06 |
| AUCSeg (Ours) | **49.20** | **80.59** | 59.45 | **45.52** | **82.71** | 93.91 | 72.72 | 81.67 | **42.73** | 56.95 | 41.93 | **40.72** |

metrics, AUCSeg does not achieve the best performance. Even in these cases, AUCSeg still secures the runner-up status. We analyze the performance trade-off between head and tail classes in Appendix H. These experimental results underscore the effectiveness of our proposed method. We further present the results for each tail class in Appendix G.1, and analyze the reasons for the varying performance improvements of tail classes across the three datasets in Appendix G.2.

Figure 4 displays the qualitative results on the Cityscapes validation set. Benefiting from our proposed AUC and T-Memory Bank techniques, AUCSeg segments objects in tail classes more accurately. It correctly distinguishes between bicycles and motorcycles and successfully identifies distant traffic lights, which other methods overlook. More qualitative results can be found in Appendix G.3.

## 5.3 Backbone Extension

In Section 5.1, we select the current SOTA Seg-NeXt as the backbone. Nevertheless, AUC-Seg can also adapt to other backbones, consistently delivering effective results. Table 2 presents the experimental results of AUCSeg when using DeepLabV3+, EMANet, OCRNet, and ISANet as backbones. These results reveal significant improvements in both overall mIoU and tail classes mIoU with AUCSeg. Notably, on ISANet, the increases are 5.54% and 6.49%. Moreover, AUCSeg enhances performance across various model sizes and different pixel-level long-tail problems, as detailed in Appendix G.4 and Appendix G.4. **This demonstrates the superiority of our proposed AUC-Seg for long-tailed semantic segmentation.**

Table 2: Results of AUCSeg using different backbones in terms of mIoU (%).

| Backbone | AUCSeg | Overall | Tail |
|---|---|---|---|
| DeepLabV3+ | ✗ | 31.95 | 26.01 |
| | ✓ | **36.13** | **31.10** |
| EMANet | ✗ | 32.83 | 27.36 |
| | ✓ | **36.32** | **31.39** |
| OCRNet | ✗ | 29.64 | 23.72 |
| | ✓ | **34.82** | **29.75** |
| ISANet | ✗ | 29.53 | 23.64 |
| | ✓ | **35.07** | **30.13** |

## 5.4 Ablation studies

We perform several ablation studies to test the effectiveness of different modules and hyperparameters. All experiments are conducted on the **ADE20K** validation set.

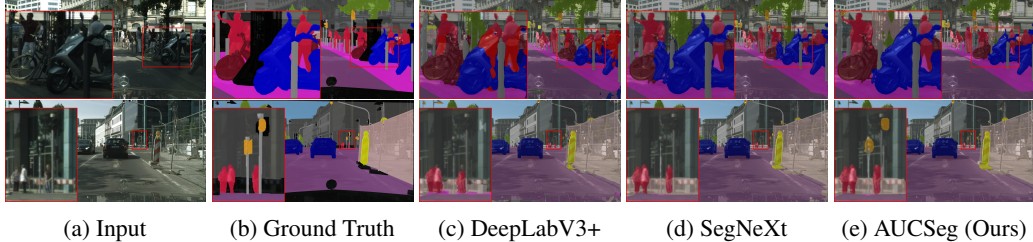

| (a) Input | (b) Ground Truth | (c) DeepLabV3+ | (d) SegNeXt | (e) AUCSeg (Ours) |

Figure 4: Qualitative results on the **Cityscapes** val set. Red rectangles highlight and magnify the image details in the lower left corner.

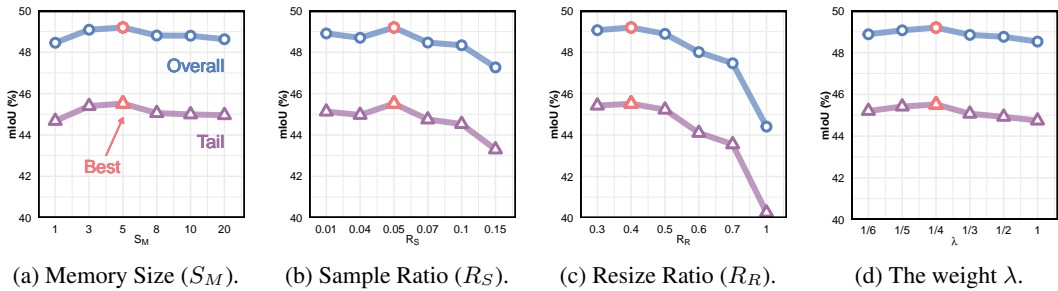

| (a) Memory Size ($S_M$). | (b) Sample Ratio ($R_S$). | (c) Resize Ratio ($R_R$). | (d) The weight $\lambda$. |

Figure 5: Ablation Study on Hyper-Parameters.

**The Effectiveness of AUC Optimization and T-Memory Bank.** Table 3 details our step-by-step ablation study on the AUC and T-Memory Bank components of AUCSeg. Compared to the baseline SegNeXt, AUC enhances performance by $+1.01\%$ and $+1.42\%$ in overall and tail classes. The T-Memory Bank further addresses the imbalance issue, yielding improvements of $+1.75\%$ overall and $+2.24\%$ in tail classes. Our results also show that employing T-Memory Bank without AUC yields no significant improvements, underscoring the necessity of AUC Loss. More ablation experiments on the effectiveness of AUC optimization and TMB are deferred to Appendix G.6, G.7, G.8, and G.9.

Table 3: Ablation study on the effectiveness of AUC Optimization and T-Memory Bank (TMB) in terms of mIoU (%).

| Model | AUC | TMB | Overall | Tail |
|---|---|---|---|---|
| SegNeXt | | | 47.45 | 43.28 |
| SegNeXt+AUC | ✓ | | 48.46 | 44.70 |
| SegNeXt+TMB | | ✓ | 47.86 | 43.86 |
| AUCSeg | ✓ | ✓ | **49.20** | **45.52** |

**Ablation Study on Hyper-Parameters.** Figure 5a ablates the maximum number of images stored per class in the Memory Branch, referred to as Memory Size ($S_M$). For the ADE20K dataset, optimal performance occurs when $S_M = 5$, and performance shows little sensitivity to changes in $S_M$. Figure 5b ablates the Sample Ratio ($R_S$), the fraction of classes sampled from the Memory Branch relative to the total number of missing tail classes. Figure 5c ablates the Resize Ratio ($R_R$), the scaling factor for the sampled pixels. For ADE20K, the best result is obtained when $R_S = 0.05$ and $R_R = 0.4$. A potential reason for their small value is that the original image is overwritten when $R_S$ and $R_R$ are too large, resulting in poor training performance. Figure 5d ablates the weight $\lambda$ for $\ell_{ce}$ and $\ell_{auc}$, with $\lambda = \frac{1}{4}$ providing slightly better results. The influence of $\lambda$ on performance is minimal. Detailed results from this hyper-parameter ablation study are available in Appendix G.10 and G.11.

## 6 Conclusion

This paper explores AUC optimization in the context of PLSS tasks. To begin with, we theoretically study the generalization performance of AUC-oriented PLSS by overcoming the two-layer coupling issue across the loss terms of AUCSeg therein. The corresponding results show that applying AUC optimization to PLSS could also enjoy a promising performance. Subsequently, we propose a novel T-Memory Bank to reduce the significant memory demand for the mini-batch optimization of AUCSeg. Finally, comprehensive experiments suggest the effectiveness of our proposed AUCSeg.

# Acknowledgments

This work was supported in part by the National Key R&D Program of China under Grant 2018AAA0102000, in part by National Natural Science Foundation of China: 62236008, U21B2038, U23B2051, 61931008, 62122075, 92370102, 62406305, 62471013 and 62476068, in part by Youth Innovation Promotion Association CAS, in part by the Strategic Priority Research Program of the Chinese Academy of Sciences, Grant No. XDB0680000, in part by the Innovation Funding of ICT, CAS under Grant No.E000000, in part by the Postdoctoral Fellowship Program of CPSF under Grant GZB20240729 and GZB20230732, and in part by the China Postdoctoral Science Foundation under Grant No.2023M743441.

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

# Contents

# A  Symbol Definitions

In this section, Table 4 includes a summary of key notations and descriptions in this work.

Table 4: A summary of key notations and descriptions in this work.

| Notations | Descriptions |
|---|---|
| $\mathcal{D}$ | Training dataset. |
| $H/W$ | The height/width of the images in dataset $\mathcal{D}$. |
| $N$ | The number of image samples in dataset $\mathcal{D}$. |
| $K$ | The total number of classes in dataset $\mathcal{D}$. |
| $(\mathbf{X}^i, \mathbf{Y}^i)$ | The $i$-th sample and its ground-truth in dataset $\mathcal{D}$, where $i \in [1, n]$, $\mathbf{X}^i \in \mathbb{R}^{H \times W \times 3}$, $\mathbf{Y}^i \in \mathbb{R}^{H \times W \times K}$. |
| $f_\theta$ | The semantic segmentation model, where $\theta$ is its parameter. |
| $f_\theta^e / f_\theta^d$ | The encoder/decoder in the semantic segmentation model $f_\theta$. |
| $\hat{\mathbf{Y}}^i$ | The prediction of model $f_\theta$, where $\hat{\mathbf{Y}}^i = f_\theta(\mathbf{X}^i) \in \mathbb{R}^{H \times W \times K}$. |
| $(\mathbf{X}_{u,v}^i, \mathbf{Y}_{u,v}^i / \hat{\mathbf{Y}}_{u,v}^i)$ | The sample and its ground-truth/prediction of the $(u, v)$-th pixel of the $i$-th image. |
| $\mathbf{Y}_{u,v}^{ic} / \hat{\mathbf{Y}}_{u,v}^{ic}$ | The ground-truth one-hot encoding/prediction of pixel $(\mathbf{X}_{u,v}^i, \mathbf{Y}_{u,v}^i)$ in class $c$. |
| $\mathcal{D}^p$ | The set of all pixels in dataset $\mathcal{D}$. |
| $(\mathbf{X}_j^p, \mathbf{Y}_j^p)$ | The $j$-th element in $\mathcal{D}^p$, where $j \in [1, n \times (H-1) \times (W-1)]$. |
| $f_\theta^{(c)}$ | The continuous score function supporting class $c$. |
| $\mathcal{N}_c$ | The set of pixels with label $c$ in the set $\mathcal{D}^p$, where $\mathcal{N}_c = \{\mathbf{X}_k^p | \mathbf{Y}_k^p = c\}$. |
| $|\mathcal{A}|$ | The number of elements in set $\mathcal{A}$. |
| $\mu$ | The Lipschitz constant. |
| $\Omega(\cdot)$ | The lower bound. |
| $\mathcal{N}_c / \mathcal{T}_c$ | The set of pixels with label $c$ in the original image/those pixels stored in the T-Memory Bank. |
| $\tilde{\mathbf{X}}^p$ | The sample after replacing some pixels with tail classes pixels from the T-Memory Bank. |
| $n_t$ | The number of tail classes. |
| $\mathcal{C}_t$ | The labels of tail classes, where $\mathcal{C}_t = \{c_i\}_{i=1}^{n_t}$. |
| $\mathcal{M}$ | The Memory Branch, where $\mathcal{M} = \{\mathcal{M}_{c_1}, \ldots \mathcal{M}_{c_{n_t}}\}$. |
| $S_M$ | The memory size. |
| $R_S$ | The sample ratio. |
| $R_R$ | The resize ratio. |
| $T_{max}$ | The max iteration. |
| $N_b$ | The batch size. |

# B  Generalization Bounds and Its Proofs

## B.1  Preliminary Lemmas

**Lemma 1** (Jensen's Inequality). *If $X$ is a random variable and $\varphi$ is a convex function, then*

$$\varphi\left(\mathbb{E}\left[X\right]\right) \leq \mathbb{E}\left[\varphi\left(X\right)\right]. \tag{9}$$

**Assumption 1.** Assume that $\ell$ is $\mu$-Lipschitz continuous, that is

$$|\ell\left(t\right) - \ell\left(s\right)| \leq \mu|t - s|. \tag{10}$$

Assumption 1 is a pretty mild assumption. The square loss $\ell_{sq}\left(x\right) = \left(1 - x\right)^2$ satisfies Assumption 1.

**Lemma 2.** *The empirical Rademacher complexity of function $g$ with respect to the predictor $f$ is defined as:*

$$\hat{\mathfrak{R}}_\mathcal{F}(g) = \mathbb{E}_\sigma[\sup_{f \in \mathcal{F}} \frac{1}{N} \sum_{i=1}^{N} \sigma_i g(f^{(i)})]. \tag{11}$$

*where $\mathcal{F} \subseteq \{f : \mathcal{X} \rightarrow \mathbb{R}^K\}$ is a family of predictors, and $N$ refers to the size of the dataset, and $\sigma_i s$ are independent uniform random variables taking values in $\{-1, +1\}$. The random variables $\sigma_i$ are called Rademacher variables.*

**Lemma 3.** *Let $\mathbb{E}[g]$ and $\hat{\mathbb{E}}[g]$ represent the expected risk and empirical risk, and $\mathcal{F} \subseteq \{f : \mathcal{X} \to \mathbb{R}^K\}$. Then with probability at least $1 - \delta$ over the draw of an i.i.d. sample S of size n, the generalization bound holds:*

$$\sup_{f \in \mathcal{F}} \left( \mathbb{E}[g(f)] - \hat{\mathbb{E}}[g(f)] \right) \leq 2\hat{\mathfrak{R}}_{\mathcal{F}}(g) + 3\sqrt{\frac{\log \frac{2}{\delta}}{2n}}. \tag{12}$$

**Definition 1** (Fractional Independent Vertex Cover, and Fractional Chromatic Number [103]). Let a graph be $G = (V, E)$.

(1) A **fractional vertex cover** of $G$ is a family $\{(F_j, \omega_j)\}_j$ of pairs $(F_j, \omega_j)$, where $F_j \subseteq V(G)$, $\omega_j \in (0, 1]$, and $\sum_{j:v \in F_j} \omega_j = 1, \forall v \in V(G)$.

(2) An **independent set** of $G$ is a set of vertices in $G$ with no two adjacent. Let $\mathcal{I}(G)$ denote the set of independent sets of $G$.

(3) A fractional vertex cover is a **fractional independent vertex cover** $\{(I_j, \omega_j)\}_j$ of $G$ if $\forall j$, $I_j \in \mathcal{I}(G)$.

(4) A **fractional coloring** of a graph $G$ is a mapping $g : \mathcal{I}(G) \to (0, 1]$ such that $\sum_{I \in \mathcal{I}(G):v \in I} g(I) \geq 1, \forall v \in V(G)$. The **fractional chromatic number** $\chi_f(G)$ is the minimum of the value $\sum_{I \in \mathcal{I}(G)} g(I)$ over fractional colorings of $G$.

Notably, the minimum of $\sum_j \omega_j$ over all fractional independent vertex covers $\{(I_j, \omega_j)\}_j$ of $G$ is the fractional chromatic number $\chi_f(G)$.

**Definition 2** (Dependency Graph [44]). An (undirected) graph $G = (V, E)$ is called a dependency graph associated with a random vector (or random variables) $\mathbf{x} = (x_1, \dots, x_m)$ if

(1) $V(G) = [m]$.

(2) For all disjoint vertex sets $I, J \subseteq [m]$, if $I, J$ are not adjacent in $G$, then random variables $\{x_i\}_{i \in I}$ and $\{x_j\}_{j \in J}$ are independent.

A useful result is Janson's decomposition property [44], which combines the concept of dependency graphs with fractional independent vertex covers. The property states that if interdependent random variables $(x_i)_{i \in [m]}$ is associated with a dependency graph $G$ with a fractional independent vertex cover $(I_j, \omega_j)_{j \in [J]}$, then, the sum of the interdependent variables, can be equivalently decomposed into a weighted sum of sums of independent variables:

$$\sum_{i=1}^{m} x_i = \sum_{i=1}^{m} \sum_{j=1}^{J} \omega_j \mathbf{1}_{i \in I_j} x_i = \sum_{j=1}^{J} \omega_j \sum_{i \in I_j} x_i. \tag{13}$$

### B.2 Key Lemmas

**Lemma 4.** *Let $\mathcal{D}$ represents the training set, and $+/-$ respectively denote the categories $c/c'$. The function $f_{i,j}^{(+)}$ represents the score function for the $(i, j)$-th pixel in category c. For a set $\mathcal{A}$, $|\mathcal{A}|$ denotes the number of elements in the set. We have:*

$$\begin{aligned}
\mathbb{E}_{\mathcal{D}} \left[ \ell_{auc}^{c,c'} \right] &= \mathbb{E}_{\mathcal{D}} \left[ \sum_{\boldsymbol{x}_m^P \in \mathcal{N}_c} \sum_{\boldsymbol{x}_n^P \in \mathcal{N}_{c'}} \frac{1}{|\mathcal{N}_c| |\mathcal{N}_{c'}|} \ell_{sq}^{c,c',m,n} \Big| c, c' \right] \\
&= \mathop{\mathbb{E}}_{\mathbf{X}_1 \sim \mathcal{D}} \left[ \tilde{\ell}_{+,-}^{inner} \right] + \mathop{\mathbb{E}}_{\mathbf{X}_1, \mathbf{X}_2 \sim \mathcal{D}} \left[ \tilde{\ell}_{+,-}^{inter} \right].
\end{aligned} \tag{14}$$

*where,*

$$\tilde{\ell}_{+,-}^{inner} = \frac{|\mathcal{D}|\, n\left(\mathbf{X}_1^+\right) n\left(\mathbf{X}_1^-\right)}{|\mathcal{N}_+|\,|\mathcal{N}_-|} \sum_{\substack{(i_1,j_1)\in\mathbf{X}_1^+ \\ (i_2,j_2)\in\mathbf{X}_1^-}} \frac{\ell_{sq}\left(\tilde{f}_{i_1,j_1}^{(+)}, \tilde{f}_{i_2,j_2}^{(+)}, \mathbf{X}, \mathbf{Y}\right)}{|\mathcal{D}|\, n\left(\mathbf{X}_1^+\right) n\left(\mathbf{X}_1^-\right)},$$

$$\tilde{\ell}_{+,-}^{inter} = \frac{|\mathcal{D}|\,(|\mathcal{D}|-1) n\left(\mathbf{X}_1^+\right) n\left(\mathbf{X}_2^-\right)}{|\mathcal{N}_+|\,|\mathcal{N}_-|} \sum_{\substack{(i_1,j_1)\in\mathbf{X}_1^+ \\ (i_2,j_2)\in\mathbf{X}_2^-}} \frac{\ell_{sq}\left(\tilde{f}_{i_1,j_1}^{(+)}, \tilde{f}_{i_2,j_2}^{(+)}, \mathbf{X}, \mathbf{Y}\right)}{|\mathcal{D}|\,(|\mathcal{D}|-1) n\left(\mathbf{X}_1^+\right) n\left(\mathbf{X}_2^-\right)},$$

*and $n\left(\mathbf{X}^+\right)/n\left(\mathbf{X}^-\right)$ represents the number of positive/negative samples in image $\mathbf{X}$.*

**Definition 3** (Covering Number [110, 54])**.** Let $\mathcal{F}$ be class of real-valued fucntions, defined over a space $\mathcal{Z}$ and $S := \left\{(\mathbf{X}_1, \mathbf{Y}_1), \ldots, (\mathbf{X}_n, \mathbf{Y}_n)\right\} \in \mathcal{Z}^n$ of cardinality $n$. For any $\epsilon > 0$, the empirical $\ell_\infty$-norm covering number $\mathcal{N}\left(\mathcal{F}, ||\cdot||_\infty, S, \epsilon\right)$ w.r.t $S$ is defined as the minimal number $m$ of a collection of vectors $\mathbf{v}^1, \ldots, \mathbf{v}^m \in \mathbb{R}^n$ such that ($\mathbf{v}_i^j$ is the $i$-th component of the vector $\mathbf{v}^j$)

$$\sup_{f\in\mathcal{F}} \min_{j=1,\ldots,m} \max_{i=1,\ldots,n} \left| f\left(\mathbf{X}_i, \mathbf{Y}_i\right) - \mathbf{v}_i^j \right| \leq \epsilon. \tag{15}$$

In this case, we call $\left\{\mathbf{v}^1, \ldots, \mathbf{v}^m\right\}$ an $(\epsilon, \ell_\infty)$-cover of $\mathcal{F}$ w.r.t $S$.

**Lemma 5.** *Let $P(\mathbf{X})$ represents the pixels in image $\mathbf{X}$, $\mathcal{F} = \left\{\left\{f_{i,j}^{(+)}\right\} : (i,j) \in P(\mathbf{X}), (\mathbf{X}, \mathbf{Y}) \in S\right\}$ and $\ell \circ \mathcal{F} = \left\{\ell \circ \left\{f_{i,j}^{(+)}\right\} : \left\{f_{i,j}^{(+)}\right\} \in \mathcal{F}\right\}$. According to Definition 3, we have:*

$$\mathcal{N}\left(\ell\circ\mathcal{F}, ||\cdot||_\infty, S, \epsilon\right) \leq \mathcal{N}\left(\mathcal{F}, ||\cdot||_\infty, S, \epsilon/2\mu\rho_{\max}\right), \tag{16}$$

*where $\rho_{\max} = \max_{\mathbf{X}} \frac{|\mathcal{D}|n\left(\mathbf{X}^+\right)n\left(\mathbf{X}^-\right)}{|\mathcal{N}_+||\mathcal{N}_-|}$.*

*Proof.* For any $g = \ell \circ f$, we can find $\tilde{g} = \ell \circ \tilde{f}$ that satisfies the following conditions:

$$||g - \tilde{g}||_{\infty,S} = \max_{(\mathbf{X},\mathbf{Y})\in S} |g(\mathbf{X}, \mathbf{Y}) - \tilde{g}(\mathbf{X}, \mathbf{Y})|$$

$$\leq \max_{(\mathbf{X},\mathbf{Y})\in S} \left| \frac{|\mathcal{D}|\, n\left(\mathbf{X}^+\right) n\left(\mathbf{X}^-\right)}{|\mathcal{N}_+|\,|\mathcal{N}_-|} \sum_{\substack{(i_1,j_1)\in\mathbf{X}^+ \\ (i_2,j_2)\in\mathbf{X}^-}} \frac{1}{n\left(\mathbf{X}^+\right) n\left(\mathbf{X}^-\right)} \left[\ell_{sq} - \tilde{\ell}_{sq}\right] \right| \tag{17}$$

$$\leq \max_{(\mathbf{X},\mathbf{Y})\in S} \frac{|\mathcal{D}|\, n\left(\mathbf{X}^+\right) n\left(\mathbf{X}^-\right)}{|\mathcal{N}_+|\,|\mathcal{N}_-|} \sum_{\substack{(i_1,j_1)\in\mathbf{X}^+ \\ (i_2,j_2)\in\mathbf{X}^-}} \frac{1}{n\left(\mathbf{X}^+\right) n\left(\mathbf{X}^-\right)} \left|\ell_{sq} - \tilde{\ell}_{sq}\right|,$$

where

$$\ell_{sq} - \tilde{\ell}_{sq} = \ell_{sq}\left(f_{i_1,j_1}^{(+)}, f_{i_2,j_2}^{(+)}, \mathbf{X}, \mathbf{Y}\right) - \ell_{sq}\left(\tilde{f}_{i_1,j_1}^{(+)}, \tilde{f}_{i_2,j_2}^{(+)}, \mathbf{X}, \mathbf{Y}\right).$$

According to Assumption 1, we have:

$$\left| \ell_{sq}\left(f_{i_1,j_1}^{(+)}, f_{i_2,j_2}^{(+)}, \mathbf{X}, \mathbf{Y}\right) - \ell_{sq}\left(\tilde{f}_{i_1,j_1}^{(+)}, \tilde{f}_{i_2,j_2}^{(+)}, \mathbf{X}, \mathbf{Y}\right) \right|$$
$$\leq \mu \left| \left(f_{i_1,j_1}^{(+)}(\mathbf{X}, \mathbf{Y}) - f_{i_2,j_2}^{(+)}(\mathbf{X}, \mathbf{Y})\right) - \left(\tilde{f}_{i_1,j_1}^{(+)}(\mathbf{X}, \mathbf{Y}) - \tilde{f}_{i_2,j_2}^{(+)}(\mathbf{X}, \mathbf{Y})\right) \right|$$
$$= \mu \left| \left(f_{i_1,j_1}^{(+)}(\mathbf{X}, \mathbf{Y}) - \tilde{f}_{i_1,j_1}^{(+)}(\mathbf{X}, \mathbf{Y})\right) + \left(\tilde{f}_{i_2,j_2}^{(+)}(\mathbf{X}, \mathbf{Y}) - f_{i_2,j_2}^{(+)}(\mathbf{X}, \mathbf{Y})\right) \right| \tag{18}$$
$$\leq \mu \left[ \left| f_{i_1,j_1}^{(+)}(\mathbf{X}, \mathbf{Y}) - \tilde{f}_{i_1,j_1}^{(+)}(\mathbf{X}, \mathbf{Y}) \right| + \left| \tilde{f}_{i_2,j_2}^{(+)}(\mathbf{X}, \mathbf{Y}) - f_{i_2,j_2}^{(+)}(\mathbf{X}, \mathbf{Y}) \right| \right]$$
$$= \mu \left[ \left| f_{i_1,j_1}^{(+)}(\mathbf{X}, \mathbf{Y}) - \tilde{f}_{i_1,j_1}^{(+)}(\mathbf{X}, \mathbf{Y}) \right| + \left| f_{i_2,j_2}^{(+)}(\mathbf{X}, \mathbf{Y}) - \tilde{f}_{i_2,j_2}^{(+)}(\mathbf{X}, \mathbf{Y}) \right| \right].$$

Denote $\rho(x)$ by $\frac{|\mathcal{D}|n(\mathbf{X}^+)n(\mathbf{X}^-)}{|\mathcal{N}_+||\mathcal{N}_-|}$. Therefore,

$$
\begin{aligned}
||g - \tilde{g}||_{\infty,S} &\le 2\mu \max_{(\mathbf{X},\mathbf{Y})\in S} \max_{(i,j)\in\mathbf{X}} \rho(x) \left| f_{i,j}^{(+)}(\mathbf{X},\mathbf{Y}) - \tilde{f}_{i,j}^{(+)}(\mathbf{X},\mathbf{Y}) \right| \\
&\le 2\mu\rho_{\max} \max_{(\mathbf{X},\mathbf{Y})\in S} \max_{(i,j)\in\mathbf{X}} \left| f_{i,j}^{(+)}(\mathbf{X},\mathbf{Y}) - \tilde{f}_{i,j}^{(+)}(\mathbf{X},\mathbf{Y}) \right| \\
&= 2\mu\rho_{\max}||f - \tilde{f}||_{\infty,S},
\end{aligned}
\tag{19}
$$

where $\rho_{\max} \triangleq \max_{\mathbf{X}} \rho(\mathbf{X})$.

Define a $\frac{\epsilon}{2\mu\rho_{\max}}$-covering of the class $\mathcal{F}$ with $||\cdot||_\infty$ norm:

$$
\{\mathcal{C}_1,\ldots,\mathcal{C}_N\},
$$

with

$$
N = \mathcal{N}(\mathcal{F}, ||\cdot||_\infty, S, \epsilon/2\mu\rho_{\max}).
\tag{20}
$$

There exists a $f^{\mathcal{C}_k} = \left\{\left\{f_{i,j}^{\mathcal{C}_k}\right\} : (i,j) \in P(\mathbf{X})\right\}$, such that for any $f = \left\{\left\{f_{i,j}^{(+)}\right\} : (i,j) \in P(\mathbf{X})\right\} \in \mathcal{C}_k \cap \mathcal{F}$:

$$
\max_{(i,j)\in P(\mathbf{X})} |f_{i,j}^{(+)} - f_{i,j}^{\mathcal{C}_k}| \le \frac{\epsilon}{2\mu\rho_{\max}},
\tag{21}
$$

which implies that

$$
\max_{(i,j)\in P(\mathbf{X})} |g_f - g_{f^{\mathcal{C}_k}}| \le 2\mu\rho_{\max} \cdot \frac{\epsilon}{2\mu\rho_{\max}} = \epsilon,
\tag{22}
$$

where $g_f = \ell \circ f$ and $g_{f^{\mathcal{C}_k}} = \ell \circ f^{\mathcal{C}_k}$.

Denote

$$
\mathcal{C}_{g,i} = \left\{ g_f : \max_{(i,j)\in P(\mathbf{X})} |g_f - g_{f^{\mathcal{C}_k}}| \le \epsilon \right\},
\tag{23}
$$

then $\{\mathcal{C}_{g,1},\ldots,\mathcal{C}_{g,N}\}$ realizes an $\epsilon$-covering of $\ell \circ f$. Hence, the minimum size of the $\epsilon$-covering is at most $N$. Mathematically, we then have:

$$
\mathcal{N}(\ell \circ \mathcal{F}, ||\cdot||_\infty, S, \epsilon) \le \mathcal{N}(\mathcal{F}, ||\cdot||_\infty, S, \epsilon/2\mu\rho_{\max}).
\tag{24}
$$

This completed the proof. $\square$

**Lemma 6** ([52]). *Let $\mathcal{F}$ be a real-valued function class taking values in $[0,1]$, and assume that $0 \in \mathcal{F}$. Let $S$ be a finite sample of size $n$. For any $2 \le p \le \infty$, we have the following relationship between the Rademacher complexity $\hat{\mathfrak{R}}(\mathcal{F})$ and the covering number $\mathcal{N}(\mathcal{F}, ||\cdot||_p, S, \epsilon)$.*

$$
\hat{\mathfrak{R}}(\mathcal{F}) \le \inf_{\alpha>0}\left(4\alpha + \frac{12}{\sqrt{n}}\int_\alpha^1 \sqrt{\log\mathcal{N}(\mathcal{F}, ||\cdot||_p, S, \epsilon)}d\epsilon\right).
\tag{25}
$$

**Lemma 7** ([72, 3]). *Given a sample $\tilde{\mathcal{D}} = \{(\tilde{x}_i, \tilde{y}_i)\}_{i=1}^m$ where $\tilde{x}_i \in \tilde{\mathcal{X}}$, $\tilde{y}_i \in \tilde{\mathcal{Y}}$ and $\tilde{\mathcal{D}}$ is associated with a dependency graph $G$, where $\chi_f(G)$ is its fractional chromatic number, and a loss function $L : \tilde{\mathcal{X}} \times \tilde{\mathcal{Y}} \times \tilde{\mathcal{F}} \to [0,M]$, where $\tilde{\mathcal{F}} = \{\tilde{f} : \tilde{\mathcal{X}} \to \mathbb{R}\}$. Then, for any $\delta \in (0,1)$, the following generalization bound holds with probability at least $1 - \delta$:*

$$
\forall \tilde{f} \in \tilde{\mathcal{F}}, R(\tilde{f}) \le \hat{R}_{\tilde{D}}(\tilde{f}) + 2\hat{\mathfrak{R}}_{\tilde{D}}^*(L \circ \tilde{\mathcal{F}}) + 3M\sqrt{\frac{\chi_f(G)}{2m}\log\left(\frac{2}{\delta}\right)},
\tag{26}
$$

*where $\hat{\mathfrak{R}}_{\tilde{D}}^*(L \circ \tilde{\mathcal{F}})$ is the empirical fractional Rademacher complexity of the loss space.*

**Lemma 8.** *Given a sample $\tilde{\mathcal{D}} = \{(\tilde{x}_i, \tilde{y}_i)\}_{i=1}^{2m}$ where $\tilde{x}_i \in \tilde{\mathcal{X}}$, $\tilde{y}_i \in \tilde{\mathcal{Y}}$ and $\tilde{\mathcal{D}}$ is associated with a dependency graph $G$, where $\chi_f(G)$ is its fractional chromatic number, and each fractional independent vertex cover contains two independent samples, one positive and one negative. Under these conditions, $\chi_f(G)$ satisfies:*

$$
\chi_f(G) = 2(2m - 1)
\tag{27}
$$

*Proof.* The calculation of the fractional chromatic number can be transformed into finding how many groups can be formed where each group contains $m$ ordered pairs of positive and negative samples. In each group, each sample appears only once, and there are no duplicate ordered pairs of positive and negative samples across all groups.

For example, in a dataset where $2m = 4$, there exist 6 groups:

$$
\begin{array}{ll}
\text{Group 1: } (1,2), (3,4) & \text{Group 2: } (2,1), (4,3) \\
\text{Group 3: } (1,3), (2,4) & \text{Group 4: } (3,1), (4,2) \\
\text{Group 5: } (1,4), (2,3) & \text{Group 6: } (4,1), (3,2)
\end{array}
$$

Therefore, $\chi_f(G) = 6$.

For $2m$ samples, we can extract $A_{2m}^2$ ordered pairs of positive and negative samples. Since these samples have an equal status in the dataset, they appear the same number of times among these $A_{2m}^2$ pairs.

After selecting the first group from these $A_{2m}^2$ pairs, $A_{2m}^2 - m$ pairs of positive and negative samples remain. The frequency of each sample appearing in these remaining pairs remains equal.

We continue to select the second group, and so on, until the last group. The frequency of each sample in the remaining pairs still remains equal.

Thus, we can find $\frac{A_{2m}^2}{m}$ groups, *i.e.* , $\chi_f(G) = 2(2m - 1)$.

This completed the proof. $\qquad\square$

## B.3   Proof of the Main Result

**Restate of Theorem 1** (Generalization Bound for AUCSeg). Let $\mathbb{E}_{\mathcal{D}}\left[\hat{\mathcal{L}}_{\mathcal{D}}(f)\right]$ be the population risk of $\hat{\mathcal{L}}_{\mathcal{D}}(f)$. Assume $\mathcal{F} \subseteq \{f : \mathcal{X} \to \mathbb{R}^{H \times W \times K}\}$, where $H$ and $W$ represent the height and width of the image, and $K$ represents the number of categories, $\hat{\mathcal{L}}^{(i)}$ is the risk over $i$-th sample, and is $\mu$-Lipschitz with respect to the $l_\infty$ norm, (*i.e.* $\|\hat{\mathcal{L}}(x) - \hat{\mathcal{L}}(y)\|_\infty \le \mu \cdot \|x - y\|_\infty$). There exists three constants $A > 0$, $B > 0$ and $C > 0$, the following generalization bound holds with probability at least $1 - \delta$ over a random draw of i.i.d training data (at the image-level):

$$
\left|\hat{\mathcal{L}}_{\mathcal{D}}(f) - \mathbb{E}_{\mathcal{D}}\left[\hat{\mathcal{L}}_{\mathcal{D}}(f)\right]\right| \le \frac{8}{N} + \frac{\eta_{\text{inner}} + \eta_{\text{inter}}}{\sqrt{N}} \sqrt{A \log(2B\mu\tau Nk + C)}
$$

$$
+ 3 \left( \sqrt{\frac{1}{2N}} + K\sqrt{1 - \frac{1}{N}} \right) \sqrt{\log\left(\frac{4K(K-1)}{\delta}\right)},
$$

where

$$
\eta_{\text{inner}} = \frac{48\mu\tau \ln N}{N}, \quad \eta_{\text{inter}} = 2\sqrt{2}\tau,
$$

$$
\tau = \left( \max_{c \in [K]} \frac{n_{\max}^{(c)}}{n_{\text{mean}}^{(c)}} \right)^2,
$$

$n_{\max}^{(c)} = \max_{\mathbf{X}} n(\mathbf{X}^{(c)})$, $n_{mean}^{(c)} = \sum_{i=1}^{N} n(\mathbf{X}_i^{(c)})$, $N = |\mathcal{D}|$, $k = H \times W$ and $\mathbf{X}^{(c)}$ represents the pixel of class $c$ in image $\mathbf{X}$.

*Proof.* First, we find that the calculation of pair-wise AUC requires both positive and negative samples. These two samples can come from the same image or from two different images. Therefore,

we transform the original problem into two sub-problems:

$$
\left| \hat{\mathcal{L}}_{\mathcal{D}}\left(f\right) - \mathbb{E}_{\mathcal{D}}\left[ \hat{\mathcal{L}}_{\mathcal{D}}\left(f\right)\right]\right|
$$

$$
= \sup_{f\in\mathcal{F}}\left[\frac{1}{K(K-1)}\sum_{c,c'}\ell_{auc}^{c,c'} - \frac{1}{K(K-1)}\sum_{c,c'}\mathbb{E}_{\mathcal{D}}\left[\ell_{auc}^{c,c'}\right]\right]
$$

$$
\overset{(Lem.1)}{\leq}\frac{1}{K(K-1)}\sum_{c,c'}\left[\sup_{f\in\mathcal{F}}\left(\ell_{auc}^{c,c'} - \mathbb{E}_{\mathcal{D}}\left[\ell_{auc}^{c,c'}\right]\right)\right] \tag{28}
$$

$$
\overset{(Lem.4)}{=}\frac{1}{K(K-1)}\sum_{c,c'}\left[\underbrace{\sup_{f\in\mathcal{F}}\left(\ell_{+,-}^{\text{inner}} - \mathbb{E}_{\mathcal{D}}\left[\tilde{\ell}_{+,-}^{\text{inner}}\right]\right)}_{\text{\color{red}Part 1}} + \underbrace{\sup_{f\in\mathcal{F}}\left(\ell_{+,-}^{\text{inter}} - \mathbb{E}_{\mathcal{D}}\left[\tilde{\ell}_{+,-}^{\text{inter}}\right]\right)}_{\text{\color{red}Part 2}}\right],
$$

where $+/-$ respectively denote the categories $c/c'$.

For Part 1, we use the complexity measure technique of the covering number.

Assuming $\log\mathcal{N}\left(\mathcal{F},||\cdot||_{\infty},S,\epsilon\right)\leq\frac{A}{\epsilon^2}\log\left[\frac{B}{\epsilon}\cdot|\mathcal{D}|\cdot k + C\right]$, where $k = H\times W$ is the pixel count in an image. $A$, $B$ and $C$ represent constants. Based on Lemma 5, we have:

$$
\begin{aligned}
\log\mathcal{N}\left(\ell\circ\mathcal{F},||\cdot||_{\infty},S,\epsilon\right) &\leq \log\mathcal{N}\left(\mathcal{F},||\cdot||_{\infty},S,\epsilon/2\mu\rho_{\max}\right)\\
&\leq \frac{4A\mu^2\rho_{\max}^2}{\epsilon^2}\log\left[\frac{2B\mu\rho_{\max}}{\epsilon}\cdot|\mathcal{D}|\cdot k + C\right],
\end{aligned} \tag{29}
$$

$$
\begin{aligned}
\rho_{\max} &= \max_{\mathbf{X}}\frac{|\mathcal{D}|\,n\left(\mathbf{X}^+\right)n\left(\mathbf{X}^-\right)}{|\mathcal{N}_+|\,|\mathcal{N}_-|}\\
&= \frac{1}{|\mathcal{D}|}\cdot\frac{n_{\max}^+\cdot n_{\max}^-}{n_{\text{mean}}^+\cdot n_{\text{mean}}^-}\\
&\leq \frac{1}{|\mathcal{D}|}\cdot\tau
\end{aligned}
$$

where:

$$
\begin{aligned}
n_{\max}^+ &= \max_{\mathbf{X}\in\mathcal{X}}n(\mathbf{X}^+),\\
n_{\max}^- &= \max_{\mathbf{X}\in\mathcal{X}}n(\mathbf{X}^-),\\
n_{mean}^+ &= \sum_{i=1}^{|D|}n(\mathbf{X}_i^+),\\
n_{mean}^- &= \sum_{i=1}^{|D|}n(\mathbf{X}_i^-),
\end{aligned}
$$

and $\tau = \left(\max_{c\in[K]}\frac{n_{\max}^{(c)}}{n_{\text{mean}}^{(c)}}\right)^2$.

Denoted by $a := 4A\mu^2\rho_{\max}^2$, $b := 2B\mu\rho_{\max}|\mathcal{D}|\,k$ and $c := C$. Based on Lemma 6 and Lemma A.3 in [54], we have:

$$
\begin{aligned}
\hat{\mathfrak{R}}\left(\ell \circ \mathcal{F}\right) &\leq \inf_{\alpha>0}\left(4\alpha + \frac{12}{\sqrt{|\mathcal{D}|}}\int_{\alpha}^{1}\sqrt{\log\mathcal{N}\left(\ell \circ \mathcal{F}, ||\cdot||_{\infty}, S, \epsilon\right)}d\epsilon\right) \\
&\leq \inf_{\alpha>0}\left(4\alpha + \frac{12}{\sqrt{|\mathcal{D}|}}\int_{\alpha}^{1}\frac{\sqrt{a\log\left(b/\epsilon+c\right)}}{\epsilon}d\epsilon\right) \\
&\leq \frac{4}{|\mathcal{D}|} + \frac{12}{\sqrt{|\mathcal{D}|}}\int_{1/|\mathcal{D}|}^{1}\frac{\sqrt{a\log\left(b\,|\mathcal{D}|+c\right)}}{\epsilon}d\epsilon \\
&= \frac{4}{|\mathcal{D}|} + \frac{12\ln|\mathcal{D}|}{\sqrt{|\mathcal{D}|}}\sqrt{a\log\left(b\,|\mathcal{D}|+c\right)}.
\end{aligned}
\tag{30}
$$

Substituting this result into Lemma 3, with probability at least $1-\delta$, we have

$$
\begin{aligned}
&\sup_{f\in\mathcal{F}}\left(\ell_{+,-}^{\text{inner}} - \mathbb{E}_{\mathcal{D}}\left[\tilde{\ell}_{+,-}^{\text{inner}}\right]\right) \\
&\leq \frac{8}{|\mathcal{D}|} + \frac{48\mu\tau\ln|\mathcal{D}|}{|\mathcal{D}|^{1.5}}\sqrt{A\log\left(2B\mu\tau\,|\mathcal{D}|\,k+C\right)} + 3\sqrt{\frac{\log\left(\frac{4K(K-1)}{\delta}\right)}{2\,|\mathcal{D}|}}.
\end{aligned}
\tag{31}
$$

For Part 2, we use the complexity measure technique of the fractional chromatic number and covering number.

According to Lemma 7, calculating the generalization bound only requires knowing the chromatic complexity. Based on Equation (13), we have

$$
\hat{\mathfrak{R}}\left(\ell \circ \mathcal{F}\right) = \frac{1}{|\mathcal{D}|\left(|\mathcal{D}|-1\right)}\mathbb{E}_{\sigma}\left[\sum_{j\in[J]}w_j\left(\sup_{f\in\mathcal{F}}\sum_{\substack{(i_1,j_1)\in I_j \\ (i_2,j_2)\in I_j}}\sigma_{i,j}^{1,2}\rho(x)\ell_{sq}\left(\tilde{f}_{i_1,j_1}^{(+)}, \tilde{f}_{i_2,j_2}^{(+)}, \mathbf{X}, \mathbf{Y}\right)\right)\right],
\tag{32}
$$

where $\rho(x) = \frac{|\mathcal{D}|(|\mathcal{D}|-1)n(\mathbf{X}^+)n(\mathbf{X}^-)}{|\mathcal{N}_+||\mathcal{N}_-|}$, and $\omega_j$ denotes the weight assigned to the subset in the fractional vertex cover.

Similar to Part 1, using the covering number we can get

$$
\mathfrak{R}_j = \sup_{f\in\mathcal{F}}\sum_{\substack{(i_1,j_1)\in I_j \\ (i_2,j_2)\in I_j}}\sigma_{i,j}^{1,2}\rho(x)\ell_{sq}\left(\tilde{f}_{i_1,j_1}^{(+)}, \tilde{f}_{i_2,j_2}^{(+)}, \mathbf{X}, \mathbf{Y}\right) \lesssim \sqrt{c_j \cdot m_j},
\tag{33}
$$

where we define $c_j$ as $\ln^2|\mathcal{D}| \cdot A \cdot \mu^2 \cdot \rho_{\max}^2 \cdot \log\left(2B \cdot \mu \cdot \rho_{\max} \cdot |\mathcal{D}|^2 \cdot k + C\right)$ as $c_j$, and define $m_j$ as $|I_j|$.

Assume that the number of images in the dataset is even, *i.e.*, $|D| \equiv 0 \pmod{2}$. We have:

$$
\begin{aligned}
\hat{\mathfrak{R}}\left(\ell \circ \mathcal{F}\right) &= \frac{1}{|\mathcal{D}|\left(|\mathcal{D}|-1\right)} \sum_{j \in [J]} w_j \mathfrak{R}_j \\
&\lesssim \frac{1}{|\mathcal{D}|\left(|\mathcal{D}|-1\right)} \sum_{j \in [J]} w_j \sqrt{c_j \cdot m_j} \\
&= \frac{\chi_f(G)}{|\mathcal{D}|\left(|\mathcal{D}|-1\right)} \sum_{j \in [J]} \frac{w_j}{\chi_f(G)} \sqrt{c_j m_j} \\
&\leq \frac{\sqrt{\chi_f(G)}}{|\mathcal{D}|\left(|\mathcal{D}|-1\right)} \sqrt{\sum_{j \in [J]} w_j m_j c_j} \\
&= \frac{\sqrt{\chi_f(G)}}{\sqrt{|\mathcal{D}|\left(|\mathcal{D}|-1\right)}} \sqrt{\frac{\sum\limits_{j \in [J]} w_j m_j c_j}{|\mathcal{D}|\left(|\mathcal{D}|-1\right)}} \\
&\overset{(*)}{\lesssim} \frac{1}{\sqrt{|\mathcal{D}|/2}} \sqrt{\frac{\sum\limits_{j \in [J]} w_j c_j}{2(|\mathcal{D}|-1)}},
\end{aligned}
\tag{34}
$$

where $(*)$ follows the Lemma 8.

The second-order inequality is based on the fact that:

$$
\mathfrak{R}_j \lesssim \sqrt{c_j \cdot m_j},
\tag{35}
$$

define $\rho_{\max}^j$ as the largest $\rho$ in the $j$-th cluster, and that:

$$
\begin{aligned}
\sqrt{c_j} &= \rho_{\max}^j \cdot \sqrt{A \log(2B\mu\rho_{\max}|\mathcal{D}|^2 k + C)} \\
&\leq \frac{|\mathcal{D}|\left(|\mathcal{D}|-1\right) n_{\max}^+ n_{\max}^-}{|\mathcal{N}_+||\mathcal{N}_-|} \sqrt{A \log(2B\mu\rho_{\max}|\mathcal{D}|^2 k + C)} \\
&\approx \frac{n_{\max}^+ \cdot n_{\max}^-}{n_{\text{mean}}^+ \cdot n_{\text{mean}}^-} \sqrt{A \log(2B\mu\rho_{\max}|\mathcal{D}|^2 k + C)} \\
&\leq \tau \sqrt{A \log(2B\mu\tau|\mathcal{D}| k + C)}
\end{aligned}
\tag{36}
$$

where:

$$
\begin{aligned}
n_{\max}^+ &= \max_{\mathbf{X} \in \mathcal{X}} n(\mathbf{X}^+), \\
n_{\max}^- &= \max_{\mathbf{X} \in \mathcal{X}} n(\mathbf{X}^-), \\
n_{mean}^+ &= \sum_{i=1}^{|D|} n(\mathbf{X}_i^+), \\
n_{mean}^- &= \sum_{i=1}^{|D|} n(\mathbf{X}_i^-).
\end{aligned}
$$

and $\tau = \left(\max\limits_{c \in [K]} \frac{n_{\max}^{(c)}}{n_{\text{mean}}^{(c)}}\right)^2$.

Combining Equation (34) and Equation (36), we obtain:

$$
\hat{\mathfrak{R}}\left(\ell \circ \mathcal{F}\right) \leq \frac{1}{\sqrt{|\mathcal{D}|/2}} \cdot \tau \sqrt{A \log(2B\mu\tau|\mathcal{D}| k + C)}.
\tag{37}
$$

Based on Lemma 7 and Lemma 8, it comes:

$$\sup_{f \in \mathcal{F}} \left( \ell_{+,-}^{\text{inter}} - \mathbb{E}_{\mathcal{D}} \left[ \tilde{\ell}_{+,-}^{\text{inter}} \right] \right)$$

$$\leq 2\hat{\mathfrak{R}} \left( \ell \circ \mathcal{F} \right) + 3M \sqrt{\frac{\chi_f(G)}{2m} \log \left( \frac{4K(K-1)}{\delta} \right)} \tag{38}$$

$$\leq \frac{2\sqrt{2}}{\sqrt{|\mathcal{D}|}} \cdot \tau \sqrt{A \log \left( 2B\mu\tau \, |\mathcal{D}| \, k + C \right)} + 3K \sqrt{\frac{|\mathcal{D}|-1}{|\mathcal{D}|} \log \left( \frac{4K(K-1)}{\delta} \right)}.$$

Therefore, by combining Equation (28), Equation (31) and Equation (38), we can obtain:

$$\left| \hat{\mathcal{L}}_{\mathcal{D}} \left( f \right) - \mathbb{E}_{\mathcal{D}} \left[ \hat{\mathcal{L}}_{\mathcal{D}} \left( f \right) \right] \right| \leq \frac{8}{N} + \frac{\eta_{\text{inner}} + \eta_{\text{inter}}}{\sqrt{N}} \sqrt{A \log \left( 2B\mu\tau N k + C \right)}$$

$$+ 3 \left( \sqrt{\frac{1}{2N}} + K\sqrt{1 - \frac{1}{N}} \right) \sqrt{\log \left( \frac{4K(K-1)}{\delta} \right)}, \tag{39}$$

where $\eta_{\text{inner}} = \frac{48\mu\tau \ln N}{N}$, $\eta_{\text{inter}} = 2\sqrt{2}\tau$, $N = |\mathcal{D}|$ and $k = H \times W$.

This completed the proof. $\qquad\qquad\square$

## C   Proof for Propositions of Tail-class Memory Bank

**Restate of Proposition 1.**  Consider a dataset $\mathcal{D}$ that includes images with $K$ different pixel categories. Let $p_i$ represent the probability of observing a pixel with label $i$ in a given image. Randomly select $B$ images from $\mathcal{D}$ as training data, where

$$B = \Omega \left( \frac{\log(\delta/K)}{\log(1 - \min\limits_{i} p_i)} \right).$$

Then with probability at least $1 - \delta$, for any $c \in [K]$, there exists $\mathbf{X}$ in the training data that contains pixels of label $c$.

*Proof.*  Define event $A_j$ as the extraction of $N$ images where the pixels of class $j$ appear at least once. Let $p_m = \min\{p_1, p_2, \ldots, p_K\}$, where $m \in [K]$.

The probability that each category appears at least once when randomly selecting $N$ images:

$$\mathbb{P} \left( \bigcap_{i=1}^{K} A_i \right) = 1 - \mathbb{P} \left( \bigcup_{i=1}^{K} \overline{A_i} \right)$$

$$\geq 1 - \sum_{i=1}^{K} \mathbb{P} \left( \overline{A_i} \right) \tag{40}$$

$$= 1 - \sum_{i=1}^{K} (1 - p_i)^B$$

$$\geq 1 - K(1 - p_m)^B$$

When $\mathbb{P} \left( \bigcap_{i=1}^{K} A_i \right) \geq 1 - \delta$,

$$\delta \geq K(1 - p_m)^B \tag{41}$$

Therefore,

$$B = \Omega \left( \frac{\log(\delta/K)}{\log(1 - p_m)} \right) = \Omega \left( \frac{\log(\delta/K)}{\log(1 - \min\limits_{i} p_i)} \right), \tag{42}$$

This completed the proof. $\qquad\qquad\square$

# D   Details of T-Memory Bank Algorithm

In this section, Algorithm 2 provides a full version of Algorithm 1.

---

**Algorithm 2:** AUCSeg Algorithm (Full Version)

---

**Input:** Training data $\mathcal{D}$, number of tail classes $n_t$, labels of tail classes $\mathcal{C}_t = \{c_i\}_{i=1}^{n_t}$, Memory
    Branch $\mathcal{M} = \{\mathcal{M}_{c_1}, \ldots \mathcal{M}_{c_{n_t}}\}$, memory size $S_M$, sample ratio $R_S$, resize ratio $R_R$,
    max iteration $T_{max}$, batch size $N_b$

**Output:** model parameters $\theta$

1  **for** $iter = 1$ *to* $T_{max}$ **do**
2     $\mathcal{D}_B = \{(\mathbf{X}_i, \mathbf{Y}_i)\}_{i=1}^{N_b} \leftarrow SampleBatch(\mathcal{D}, N_b)$;
3     $\mathcal{C}_{miss} \subseteq \mathcal{C}_t \leftarrow MissingTailClasses(\mathcal{D}_B)$;
4     $\overline{\mathcal{C}_{miss}} = \mathcal{C}_t - \mathcal{C}_{miss}$;
5     ▷ Store Branch
6     **for** $\overline{c_m}$ *in* $\overline{\mathcal{C}_{miss}}$ **do**
7        Divide a picture containing only the $\overline{c_m}$-th class pixels from $\mathcal{D}_B$ and name it $\mathcal{P}$;
8        **if** $|\mathcal{M}_{\overline{c_m}}| < S_M$ **then**
9           Add the divided image $\mathcal{P}$ to $\mathcal{M}_{\overline{c_m}}$;
10       **else**
11          Randomly replace an image in $\mathcal{M}_{\overline{c_m}}$ with the divided image $\mathcal{P}$;
12    ▷ Retrieve Branch
13    $n_{sample} = \lceil |\mathcal{C}_{miss}| \times R_S \rceil$;
14    **for** $i = 1$ *to* $n_{sample}$ **do**
15       Randomly choose $c_m$ from $\mathcal{C}_{miss}$;
16       Remove $c_m$ in $\mathcal{C}_{miss}$;
17       **if** $|\mathcal{M}_{c_m}| \neq 0$ **then**
18          Sample from $\mathcal{M}_{c_m}$, scale according to $R_R$, paste randomly into $\mathcal{D}_B$;
19    ▷ Semantic Segmentation
20    $\hat{\mathbf{Y}}_i \leftarrow f_\theta(\mathbf{X}_i)$;
21    Calculate $\ell = \tilde{\ell}_{auc} + \lambda \ell_{ce}$ with Equation (2) and Equation (8);
22    Backpropagation updates $\theta$.

---

# E   More Discussions about T-Memory Bank

## E.1   Discussion on the Improved Version of Stratified Sampling

In this section, we introduce the definition of the improved version of stratified sampling and explain why it is not applicable to the PLSS task.

**The definition of the improved version of stratified sampling.** The improved version of stratified sampling starts by grouping images according to their categories. Due to the multi-label nature, different categories may contain the same images. After that, stratified sampling is applied to each category, making sure that even if an image is sampled more than once (as both head and tail), each mini-batch still includes at least one sample from every category.

**Reasons for inapplicability to PLSS task.** Conventional stratified sampling can hardly cover all the involved classes with a small batch size. To ensure coverage, one has to employ a much larger batch size, which results in a significantly higher computational burden. Although the improved version of stratified sampling can cover all classes, images from tail classes may appear repeatedly, leading to overfitting and consequently a degradation in performance. In contrast, our Tail-class Memory Bank only involves pasting a portion of one image onto another, effectively functioning as an implicit data augmentation. This approach mitigates the sampling problem without compromising generalization ability. The following empirical results support our assertion.

First, we counted the number of images containing pixels from each class in the Cityscapes, ADE20K, and COCO-Stuff 164K datasets. The results are shown in Table 5, Table 6, and Table 7.

Table 5: The number of images containing pixels from each class in the **Cityscapes**. Class ID represents the class number in the original dataset.

| Class ID | 5 | 0 | 2 | 8 | 13 | 1 | 7 | 10 | 11 | 6 | 9 | 18 | 4 | 12 | 3 | 17 | 14 | 15 | 16 |
|---|---|---|---|---|---|---|---|---|---|---|---|---|---|---|---|---|---|---|---|
| Num | 2949 | 2934 | 2934 | 2891 | 2832 | 2811 | 2808 | 2686 | 2343 | 1658 | 1654 | 1646 | 1296 | 1023 | 970 | 513 | 359 | 274 | 142 |

Table 6: The number of images containing pixels from each class in the **ADE20K**. Class ID represents the class number in the original dataset.

| Class ID | 1 | 4 | 3 | 5 | 6 | 2 | 13 | 9 | 16 | 18 | 7 | 23 | 15 | 20 | 21 | 37 | 12 | 11 | 44 |
|---|---|---|---|---|---|---|---|---|---|---|---|---|---|---|---|---|---|---|---|
| Num | 11588 | 9314 | 8240 | 6674 | 6579 | 6042 | 5069 | 4687 | 4266 | 3995 | 3990 | 3295 | 3276 | 3258 | 3161 | 3083 | 3061 | 2851 | 2646 |
| Class ID | 83 | 10 | 19 | 88 | 8 | 14 | 17 | 40 | 42 | 25 | 33 | 67 | 136 | 28 | 24 | 126 | 48 | 31 | 29 |
| Num | 2508 | 2421 | 2148 | 1987 | 1825 | 1791 | 1690 | 1447 | 1437 | 1404 | 1385 | 1308 | 1281 | 1229 | 1191 | 1191 | 1181 | 1172 | 1132 |
| Class ID | 68 | 135 | 94 | 99 | 58 | 54 | 39 | 43 | 65 | 35 | 149 | 22 | 34 | 139 | 26 | 70 | 27 | 113 | 90 |
| Num | 1112 | 1020 | 992 | 965 | 930 | 880 | 803 | 799 | 792 | 781 | 773 | 702 | 698 | 671 | 667 | 658 | 650 | 622 | 618 |
| Class ID | 86 | 60 | 53 | 103 | 143 | 45 | 87 | 72 | 116 | 137 | 32 | 148 | 82 | 30 | 50 | 111 | 138 | 96 | 84 |
| Num | 583 | 564 | 561 | 556 | 556 | 549 | 532 | 531 | 530 | 528 | 521 | 504 | 492 | 479 | 468 | 465 | 452 | 451 | 440 |
| Class ID | 150 | 124 | 41 | 38 | 51 | 140 | 66 | 36 | 73 | 46 | 101 | 128 | 109 | 64 | 71 | 76 | 61 | 125 | 47 |
| Num | 421 | 417 | 411 | 404 | 402 | 397 | 395 | 378 | 369 | 367 | 354 | 347 | 340 | 335 | 330 | 324 | 320 | 319 | 310 |
| Class ID | 98 | 77 | 49 | 133 | 117 | 63 | 134 | 69 | 122 | 75 | 62 | 81 | 130 | 142 | 144 | 119 | 145 | 132 | 57 |
| Num | 307 | 304 | 287 | 284 | 282 | 275 | 268 | 266 | 266 | 265 | 261 | 247 | 246 | 228 | 217 | 213 | 206 | 201 | 198 |
| Class ID | 95 | 93 | 147 | 56 | 78 | 74 | 59 | 120 | 85 | 91 | 52 | 146 | 100 | 121 | 102 | 131 | 105 | 127 | 141 |
| Num | 181 | 178 | 178 | 172 | 170 | 144 | 139 | 136 | 135 | 133 | 130 | 126 | 117 | 116 | 108 | 108 | 99 | 97 | 92 |
| Class ID | 55 | 92 | 114 | 108 | 118 | 89 | 79 | 107 | 110 | 80 | 115 | 123 | 106 | 104 | 129 | 112 | 97 | | |
| Num | 84 | 83 | 80 | 77 | 73 | 71 | 68 | 66 | 66 | 65 | 59 | 58 | 57 | 52 | 52 | 50 | 41 | | |

Table 7: The number of images containing pixels from each class in the **COCO-Stuff 164K**. Class ID represents the class number in the original dataset.

| Class ID | 0 | 157 | 145 | 160 | 93 | 84 | 112 | 120 | 161 | 128 | 111 | 153 | 137 | 169 | 155 | 56 | 2 | 60 | 118 |
|---|---|---|---|---|---|---|---|---|---|---|---|---|---|---|---|---|---|---|---|
| Num | 63965 | 36466 | 31808 | 31481 | 27657 | 23021 | 22575 | 22526 | 19095 | 18311 | 17882 | 16282 | 15402 | 14209 | 13052 | 12757 | 12238 | 11834 | 11772 |
| Class ID | 101 | 131 | 90 | 99 | 94 | 85 | 130 | 127 | 100 | 41 | 103 | 39 | 86 | 45 | 26 | 109 | 165 | 105 | 143 |
| Num | 11303 | 11137 | 10546 | 10163 | 9886 | 9849 | 9522 | 9521 | 9475 | 8910 | 8893 | 8261 | 7176 | 6782 | 6744 | 6672 | 6642 | 6618 | 6598 |
| Class ID | 116 | 106 | 114 | 7 | 13 | 24 | 164 | 73 | 133 | 159 | 147 | 97 | 170 | 123 | 89 | 142 | 71 | 95 | 144 |
| Num | 6549 | 6324 | 6252 | 6122 | 5568 | 5464 | 5290 | 5268 | 5251 | 5246 | 5114 | 5101 | 5053 | 4887 | 4858 | 4688 | 4674 | 4589 | 4589 |
| Class ID | 74 | 62 | 139 | 58 | 57 | 16 | 9 | 80 | 43 | 25 | 5 | 32 | 15 | 88 | 59 | 67 | 121 | 6 | 75 |
| Num | 4575 | 4550 | 4490 | 4450 | 4420 | 4153 | 4138 | 4135 | 3996 | 3959 | 3950 | 3879 | 3848 | 3787 | 3680 | 3677 | 3622 | 3587 | 3567 |
| Class ID | 3 | 63 | 37 | 36 | 138 | 38 | 61 | 107 | 1 | 44 | 14 | 42 | 117 | 92 | 30 | 8 | 152 | 65 | 4 |
| Num | 3500 | 3498 | 3485 | 3476 | 3397 | 3384 | 3353 | 3259 | 3241 | 3217 | 3200 | 3173 | 3169 | 3129 | 3082 | 3023 | 3016 | 3007 | 2982 |
| Class ID | 17 | 53 | 87 | 98 | 69 | 82 | 55 | 135 | 140 | 149 | 108 | 113 | 81 | 35 | 156 | 23 | 115 | 34 | 40 |
| Num | 2931 | 2925 | 2911 | 2909 | 2877 | 2813 | 2741 | 2720 | 2703 | 2667 | 2659 | 2613 | 2598 | 2585 | 2558 | 2544 | 2498 | 2494 | 2478 |
| Class ID | 166 | 27 | 28 | 72 | 162 | 136 | 168 | 33 | 29 | 20 | 46 | 110 | 66 | 77 | 134 | 48 | 163 | 158 | 132 |
| Num | 2453 | 2401 | 2387 | 2360 | 2357 | 2313 | 2297 | 2260 | 2162 | 2139 | 2130 | 2112 | 2100 | 2087 | 2068 | 2064 | 2020 | 2016 | 2009 |
| Class ID | 146 | 129 | 19 | 22 | 150 | 64 | 11 | 50 | 10 | 83 | 31 | 49 | 68 | 51 | 47 | 154 | 54 | 125 | |
| Num | 1998 | 1986 | 1962 | 1916 | 1828 | 1822 | 1732 | 1725 | 1711 | 1676 | 1652 | 1597 | 1536 | 1522 | 1509 | 1487 | 1486 | 1411 | 1405 |
| Class ID | 151 | 126 | 104 | 52 | 96 | 102 | 21 | 76 | 79 | 148 | 12 | 124 | 119 | 141 | 91 | 122 | 70 | 78 | 167 |
| Num | 1385 | 1362 | 1259 | 1134 | 1004 | 1002 | 959 | 919 | 846 | 749 | 703 | 659 | 559 | 477 | 351 | 256 | 217 | 188 | 121 |

The results indicate that images containing tail class pixels are very limited. Specifically, in the ADE20K dataset, there are only 41 images in the training set of 20210 images that contain pixels from the tail class with ID 97. As a result, tail class images are repeatedly sampled when using stratified sampling, leading to overfitting on such repeated images.

Next, we trained on the ADE20K dataset using the improved version of the stratified sampling method. The results show that, compared to using the Tail-class Memory Bank, the performance on tail classes dropped by over 3% due to heavy sample repetition in the batch. However, our T-Memory Bank, with its random pasting technique, diversifies the backgrounds of the tail classes, enabling the model to better learn the features of these tail classes.

### E.2 Discussion on Why the T-Memory Bank Works

In this section, we discuss why the primary function of the T-Memory Bank is not to enhance the diversity of tail samples.

As shown in Table 3, using a memory bank does indeed increase the diversity of tail classes as an implicit form of augmentation (comparing the rows for SegNeXt and SegNeXt+TMB in the table). However, we cannot rely solely on the bank to fully address the long-tail issue, as the bank's capacity

is always limited. This is why we also need to consider the problem from the perspective of the loss function. We find that the AUC loss focuses only on the ranking loss between positive and negative samples and is not sensitive to data distribution, fundamentally avoiding the risk of underfitting caused by insufficient training samples. **We believe that the use of the T-Memory Bank is intended to both facilitate the effectiveness of the AUC loss and enhance the diversity of tail samples.**

### E.3 Discussion on AUC and Contrastive Learning from the Perspective of the Loss Function

In this section, we compare AUC and contrastive learning from the perspective of loss functions.

**Theorem 2** (Comparison between AUC Loss and Contrastive Loss). *Minimizing the weighted contrastive loss approximately corresponds to minimizing an upper bound of the logistic AUC loss:*

$$\sum_i w_i \left[ -\log \frac{e^{f(x^i)}}{e^{f(x^i)} + \sum_{j \neq i} w_j e^{f(x^j)}} \right] \geq \sum_i \sum_{j \neq i} \frac{1}{n_i n_j} \left[ -\log \left( \frac{1}{1 + e^{f(x^j) - f(x^i)}} \right) \right],$$

*where, $w_j = \frac{1/n_j}{\sum_{k \neq i} 1/n_k}$ and $w_i = \frac{\sum_{k \neq i} 1/n_k}{n_i}$.*

*Proof.* For the AUC loss under the logistic surrogate loss function:

$$\begin{aligned}
\ell_{auc}^{logistic} &= \ell_{logistic} \left( f(x^+) - f(x^-) \right) \\
&= \sum_i \sum_{j \neq i} \frac{1}{n_i n_j} \left[ -\log \left( \frac{1}{1 + e^{f(x^j) - f(x^i)}} \right) \right] \\
&= \sum_i \sum_{j \neq i} w_i w_j \left[ -\log \left( \frac{1}{1 + e^{f(x^j) - f(x^i)}} \right) \right] \\
&= \sum_i w_i \sum_{j \neq i} w_j \left[ -\log(e^{f(x^i)}) + \log(e^{f(x^i)} + e^{f(x^j)}) \right] \\
&= \sum_i w_i \left[ -\log(e^{f(x^i)}) + \sum_{j \neq i} w_j \log(e^{f(x^i)} + e^{f(x^j)}) \right] \\
&\leq \sum_i w_i \left[ -\log(e^{f(x^i)}) + \log \left( \sum_{j \neq i} w_j (e^{f(x^i)} + e^{f(x^j)}) \right) \right] \\
&= \sum_i w_i \left[ -\log(e^{f(x^i)}) + \log \left( e^{f(x^i)} + \sum_{j \neq i} w_j e^{f(x^j)} \right) \right] \\
&= \sum_i w_i \left[ -\log \frac{e^{f(x^i)}}{e^{f(x^i)} + \sum_{j \neq i} w_j e^{f(x^j)}} \right]
\end{aligned}$$

where, $w_j = \frac{1/n_i n_j}{\sum_{k \neq i} 1/n_i n_k} = \frac{1/n_j}{\sum_{k \neq i} 1/n_k}$ and $w_i = \frac{\sum_{k \neq i} 1/n_k}{n_i}$.

This completed the proof. $\qquad\square$

The theorem indicates that minimizing a **weighted version** of contrastive loss can implicitly optimize the ovo logistic AUC loss. This paper adopts a more general form of AUC loss, in which various surrogate loss functions are explored.

# F    Additional Experimental Settings

In this section, we make a supplementation to Section 5.1.

## F.1    Datasets

We use three datasets in our experiments: Cityscapes, ADE20K, and COCO-Stuff 164K.

**Cityscapes** [19] is a dataset of urban road traffic scenes, with each image sized at $1024 \times 2048$ pixels. It consists of 5000 images with pixel-level annotations across 19 classes. The training, validation, and testing set numbers are 2975, 500, and 1525, respectively.

**ADE20K** [109] is a benchmark for scene parsing with 150 class labels. It includes over 25000 images, with 20210, 2000, and 3352 images used for training, validation, and testing, respectively.

**COCO-Stuff 164K** [9] is a large-scale dataset with $164K$ images. It is finely annotated across 171 classes.

We split the three datasets into head class, middle class, and tail class based on the proportion of pixels for each class in the training set. Table 8, Table 9, and Table 10 provide the details of these partitions.

## F.2    Implementation Details

**Network Architecture.** We perform all experiments using mmsegmentation [18] on an NVIDIA 3090 GPU. For our model, we use SegNeXt [32] as the backbone and pretrain all encoders on the ImageNet-1K [22] dataset.

**Data Augmentation.** For Cityscapes, we resize the images to $1024 \times 2048$, randomly crop them to $1024 \times 1024$, and then apply random horizontal flips. For ADE20K and COCO-Stuff 164K, the resizing is set to $512 \times 2048$, random cropping is done at $512 \times 512$, and random horizontal flips are applied as well.

**Training Strategy.** We use *Adam with Weight Decay (AdamW)* [59] optimizer with an initial learning rate of $6e$-5 and a weight decay of $0.01$. We adopt the 'poly' learning rate policy, where the initial learning rate is multiplied by $1 - \frac{iter}{max\_iter}$. Moreover, a 'linear' warmup strategy is employed at the beginning of training, allowing the learning rate to increase from $1e$-6 to the initial learning rate within 1500 iterations. The batch size is set to 2 for the Cityscapes dataset and 4 for all the other datasets. The total number of iterations is 160000 on Cityscapes and ADE20K and 80000 on COCO-Stuff 164K.

**Evaluation Metrics.** Following the setup outlined by SegNeXt [32], we conduct experiments using the *mean of Intersection over Union (mIoU)* as the evaluation metric on the validation set.

## F.3    Competitors

Here we give a more detailed summary of the competitors mentioned in the experiments.

We compared our method with 13 recent advancements and 6 long-tail approaches in semantic segmentation. The recent advancements include DeepLabV3+, EncNet, FastFCN, EMANet, DANet, HRNet, OCRNet, DNLNet, PointRend, BiSeNetV2, ISANet, STDC, and SegNeXt. The long-tail methods are VS, LA, LDAM, Focal Loss, DisAlign, and BLV, all based on SegNeXt. **To ensure fairness, we re-implement the listed methods using their publicly shared code and test them on the same hardware.**

For the semantic segmentation methods:

**DeepLabV3+** [14] combines the advantages of the spatial pyramid pooling module and the encoder-decoder structure. It explores the Xception model and applies depthwise separable convolution to both the atrous spatial pyramid pooling and decoder modules, resulting in a faster and more robust encoder-decoder network.

**EncNet** [102] enhances semantic segmentation by utilizing a context encoding module that captures global contextual information to aid in the accurate segmentation of complex scenes.

Table 8: **Cityscapes** Dataset Partition Status. The first column represents the head class, the second column represents the middle class, and the third column represents the tail class.

| Label | Pixel Ratio | Label | Pixel Ratio | Label | Pixel Ratio |
|---|---|---|---|---|---|
| road | 46.27% | fence | 0.85% | truck | 0.24% |
| building | 20.62% | person | 0.83% | bicycle | 0.24% |
| vegetation | 13.40% | terrain | 0.61% | bus | 0.24% |
| car | 6.62% | pole | 0.60% | train | 0.21% |
| sidewalk | 4.42% | wall | 0.57% | traffic light | 0.11% |
| sky | 3.63% | traffic sign | 0.40% | rider | 0.08% |
| | | | | motorcycle | 0.06% |

Table 9: **ADE20K** Dataset Partition Status. The first column represents the head class, the second column represents the middle class, and the third to sixth columns represent the tail class.

| Label | Pixel Ratio | Label | Pixel Ratio | Label | Pixel Ratio | Label | Pixel Ratio | Label | Pixel Ratio | Label | Pixel Ratio |
|---|---|---|---|---|---|---|---|---|---|---|---|
| wall | 16.93% | grass | 1.95% | sea | 0.59% | runway | 0.17% | streetlight | 0.08% | bag | 0.05% |
| building | 11.56% | cabinet | 1.95% | mirror | 0.57% | stairway | 0.17% | airplane | 0.08% | step | 0.05% |
| sky | 9.52% | sidewalk | 1.80% | seat | 0.49% | river | 0.17% | dirt | 0.08% | bicycle | 0.04% |
| floor | 6.66% | person | 1.71% | rug | 0.49% | screen | 0.17% | television | 0.08% | food | 0.04% |
| tree | 5.21% | earth | 1.61% | field | 0.48% | bridge | 0.16% | apparel | 0.07% | trade | 0.04% |
| ceiling | 4.86% | door | 1.27% | armchair | 0.48% | bookcase | 0.16% | land | 0.07% | dishwasher | 0.04% |
| road | 4.29% | table | 1.19% | fence | 0.35% | flower | 0.16% | bannister | 0.07% | tank | 0.04% |
| bed | 2.47% | mountain | 1.17% | desk | 0.34% | coffee | 0.15% | pole | 0.07% | pot | 0.04% |
| windowpane | 2.15% | curtain | 1.12% | wardrobe | 0.32% | toilet | 0.15% | bottle | 0.07% | sculpture | 0.04% |
| | | chair | 1.11% | rock | 0.32% | hill | 0.14% | stage | 0.07% | hood | 0.04% |
| | | plant | 1.09% | lamp | 0.28% | book | 0.14% | ottoman | 0.07% | vase | 0.04% |
| | | car | 1.07% | bathtub | 0.26% | blind | 0.14% | escalator | 0.07% | lake | 0.04% |
| | | water | 0.79% | railing | 0.25% | bench | 0.14% | van | 0.07% | screen | 0.04% |
| | | painting | 0.73% | base | 0.25% | palm | 0.13% | poster | 0.07% | microwave | 0.04% |
| | | sofa | 0.71% | cushion | 0.25% | countertop | 0.13% | buffet | 0.06% | sconce | 0.04% |
| | | shelf | 0.67% | box | 0.23% | kitchen | 0.13% | ship | 0.06% | animal | 0.04% |
| | | house | 0.65% | column | 0.23% | stove | 0.13% | plaything | 0.06% | tray | 0.04% |
| | | | | signboard | 0.22% | swivel | 0.11% | barrel | 0.06% | blanket | 0.04% |
| | | | | chest | 0.21% | computer | 0.11% | conveyer | 0.06% | traffic | 0.04% |
| | | | | counter | 0.20% | boat | 0.10% | fountain | 0.06% | pier | 0.04% |
| | | | | grandstand | 0.20% | arcade | 0.10% | swimming | 0.06% | shower | 0.03% |
| | | | | sink | 0.20% | hovel | 0.09% | stool | 0.06% | crt | 0.03% |
| | | | | sand | 0.20% | bus | 0.09% | canopy | 0.06% | fan | 0.03% |
| | | | | fireplace | 0.19% | bar | 0.09% | ball | 0.05% | ashcan | 0.03% |
| | | | | refrigerator | 0.19% | towel | 0.09% | waterfall | 0.05% | bulletin | 0.03% |
| | | | | skyscraper | 0.19% | truck | 0.09% | washer | 0.05% | plate | 0.03% |
| | | | | path | 0.19% | light | 0.09% | oven | 0.05% | monitor | 0.03% |
| | | | | case | 0.18% | tower | 0.08% | minibike | 0.05% | radiator | 0.03% |
| | | | | pool | 0.18% | awning | 0.08% | basket | 0.05% | clock | 0.02% |
| | | | | stairs | 0.18% | chandelier | 0.08% | tent | 0.05% | glass | 0.02% |
| | | | | pillow | 0.18% | booth | 0.08% | cradle | 0.05% | flag | 0.02% |

Table 10: **COCO-Stuff 164K** Dataset Partition Status. The first column represents the head class, the second column represents the middle class, and the third to sixth columns represent the tail class.

| Label | Pixel Ratio | Label | Pixel Ratio | Label | Pixel Ratio | Label | Pixel Ratio | Label | Pixel Ratio | Label | Pixel Ratio |
|---|---|---|---|---|---|---|---|---|---|---|---|
| person | 9.01% | fence | 0.94% | motorcycle | 0.49% | shelf | 0.28% | floor-stone | 0.16% | carrot | 0.07% |
| sky-other | 6.24% | ceiling-other | 0.93% | elephant | 0.49% | leaves | 0.28% | bird | 0.16% | parking meter | 0.07% |
| tree | 5.78% | wall-tile | 0.90% | curtain | 0.49% | gravel | 0.28% | bottle | 0.16% | traffic light | 0.07% |
| wall-concrete | 4.44% | furniture-other | 0.89% | carpet | 0.48% | wall-panel | 0.27% | bicycle | 0.16% | kite | 0.07% |
| grass | 4.23% | metal | 0.88% | cage | 0.45% | cow | 0.27% | roof | 0.15% | napkin | 0.07% |
| dining table | 3.11% | plant-other | 0.80% | water-other | 0.44% | boat | 0.27% | stone | 0.15% | skateboard | 0.06% |
| building-other | 2.98% | cabinet | 0.79% | dog | 0.44% | skyscraper | 0.26% | keyboard | 0.14% | tennis racket | 0.06% |
| road | 2.43% | train | 0.77% | house | 0.44% | wood | 0.26% | light | 0.14% | pillow | 0.06% |
| clouds | 2.39% | bus | 0.77% | paper | 0.42% | cup | 0.25% | orange | 0.14% | solid-other | 0.06% |
| sea | 2.35% | pizza | 0.74% | refrigerator | 0.38% | potted plant | 0.23% | clock | 0.13% | remote | 0.05% |
| pavement | 2.24% | ground-other | 0.73% | plastic | 0.38% | banner | 0.23% | fruit | 0.13% | salad | 0.05% |
| wall-other | 2.19% | floor-other | 0.70% | clothes | 0.37% | hill | 0.23% | hot dog | 0.13% | knife | 0.04% |
| snow | 2.04% | door-stuff | 0.67% | cake | 0.37% | cardboard | 0.23% | bridge | 0.12% | snowboard | 0.04% |
| playingfield | 1.81% | floor-tile | 0.66% | oven | 0.37% | platform | 0.23% | surfboard | 0.12% | scissors | 0.04% |
| dirt | 1.43% | wall-wood | 0.64% | laptop | 0.37% | banana | 0.23% | blanket | 0.12% | frisbee | 0.03% |
| table | 1.17% | chair | 0.64% | horse | 0.36% | wall-stone | 0.23% | fire hydrant | 0.11% | skis | 0.03% |
| bed | 1.03% | truck | 0.63% | bench | 0.36% | branch | 0.22% | stop sign | 0.11% | ceiling-tile | 0.03% |
| window-other | 1.02% | car | 0.61% | mirror-stuff | 0.36% | vegetable | 0.21% | stairs | 0.11% | tie | 0.03% |
| sand | 1.02% | bowl | 0.61% | tv | 0.34% | flower | 0.20% | apple | 0.11% | fork | 0.03% |
| | | bush | 0.60% | airplane | 0.34% | sheep | 0.19% | handbag | 0.10% | mat | 0.03% |
| | | floor-wood | 0.60% | giraffe | 0.33% | straw | 0.19% | cloth | 0.10% | spoon | 0.02% |
| | | cat | 0.60% | zebra | 0.33% | bear | 0.19% | floor-marble | 0.10% | mouse | 0.02% |
| | | couch | 0.58% | teddy bear | 0.32% | net | 0.19% | cell phone | 0.10% | baseball glove | 0.02% |
| | | textile-other | 0.57% | toilet | 0.32% | broccoli | 0.18% | cupboard | 0.10% | moss | 0.02% |
| | | wall-brick | 0.55% | counter | 0.32% | donut | 0.18% | microwave | 0.09% | toothbrush | 0.02% |
| | | river | 0.55% | rock | 0.32% | sink | 0.18% | backpack | 0.09% | baseball bat | 0.01% |
| | | fog | 0.55% | suitcase | 0.32% | book | 0.18% | wine glass | 0.09% | sports ball | 0.01% |
| | | mountain | 0.51% | desk-stuff | 0.31% | window-blind | 0.18% | tent | 0.09% | waterdrops | 0.01% |
| | | food-other | 0.50% | sandwich | 0.30% | structural-other | 0.17% | mud | 0.08% | toaster | 0.01% |
| | | | | umbrella | 0.30% | vase | 0.17% | railing | 0.08% | hair drier | 0.01% |
| | | | | railroad | 0.29% | rug | 0.17% | towel | 0.08% | | |

**FastFCN** [80] employs a novel joint upsampling module called Joint Pyramid Upsampling, which transforms the task of extracting high-resolution feature maps into a joint upsampling challenge.

**EMANet** [55] enhances semantic segmentation by utilizing an EM-based attention mechanism that iteratively refines feature representations for more accurate segmentation.

**DANet** [27] improves scene segmentation by integrating both spatial and channel-wise attention mechanisms to capture rich contextual relationships across features.

**HRNet** [71] maintains high-resolution representations through the network and progressively adds lower-resolution subnetworks to enhance the learning of spatial hierarchies, significantly improving semantic segmentation.

**OCRNet** [97] enhances semantic segmentation by leveraging object-contextual representations, which aggregates contextual information around each pixel to improve segmentation accuracy.

**DNLNet** [93] improves performance on tasks like image classification by disentangling the traditional non-local operations into two separate streams for capturing spatial and channel dependencies separately.

**PointRend** [50] introduces a novel rendering-style algorithm that selectively refines segmentation predictions at adaptively sampled points, enhancing detail accuracy in image segmentation tasks.

**BiSeNetV2** [95] utilizes an efficient architecture with inverted residuals and linear bottlenecks, enabling high-performance mobile vision applications with significantly reduced computational cost.

**ISANet** [98] improves semantic segmentation by using a novel interlaced sparse self-attention mechanism that efficiently captures long-range dependencies with fewer parameters and computational overhead.

**STDC** [25] addresses real-time semantic segmentation by proposing a novel and efficient structure that removes structural redundancy, reduces dimensions of feature maps gradually, and uses their aggregation for image representation.

**SegNeXt** [32] rethinks convolutional attention design for semantic segmentation by introducing an advanced network architecture, enhancing the model's ability to focus on relevant features for more accurate segmentation.

For the long-tail methods:

**VS** [49] proposes to leverage both multiplicative and additive logit adjustments to address label imbalance problems.

**LA** [61] advances the conventional softmax cross-entropy by ensuring Fisher consistency in minimizing the balanced error.

**LDAM** [10] improves the performance of tail classes by encouraging larger margins for tail classes.

**Focal Loss** [56] is a modified cross-entropy loss designed to address class imbalance by focusing more on hard-to-classify examples, reducing the relative loss for well-classified instances and thus boosting performance on imbalanced datasets.

**DisAlign** [104] introduces a unified framework for long-tail visual recognition by aligning feature distributions across different classes, using a novel distribution alignment technique that adjusts class-specific thresholds to mitigate the bias towards head classes and enhance recognition of tail classes.

**BLV** [77] addresses long-tailed semantic segmentation by dynamically adjusting the learning rates for the logits of different classes based on their frequency, effectively reducing the performance gap between head and tail classes.

## G    Additional Experimental Results

### G.1    Per-tail-class Results

In Figure 6, we present the results for each tail class. We select 7 classes with the fewest training samples from the Cityscapes dataset as tail classes: truck, bicycle, bus, train, traffic light, rider, and motorcycle. Our method not only achieves the highest overall mIoU but also shows significant improvements in these tail classes. Specifically, it outperforms the current SOTA method by more than 1% in several tail classes.

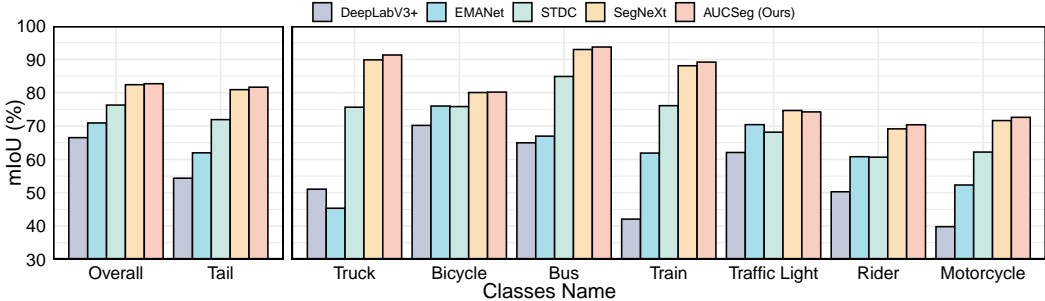

Figure 6: Per-tail-class results on **Cityscapes** val set. The tail class names are listed from left to right according to the ascending number of training samples in the dataset, with 'motorcycles' containing the fewest.

### G.2    Performance Differences Across Different Datasets

The performance gain depends on the degree of imbalance of the underlying dataset. To see this, we show the pairwise mean imbalance ratio $r_m$ (average the imbalance ratio of each class pair).

$$r_m = \frac{1}{|\mathcal{C}_h||\overline{\mathcal{C}_h}|} \sum_{a \in \mathcal{C}_h} \sum_{b \in \overline{\mathcal{C}_h}} \left(\frac{a}{b}\right) \tag{43}$$

where $\mathcal{C}_h$ represents the set containing the pixel counts of each head class, and $\overline{\mathcal{C}_h}$ denotes the set containing the pixel counts of each non-head class. The larger the $r_m$ value, the more imbalanced the dataset is.

In Table 11, we compare $r_m$ for ADE20K, Cityscapes, and COCO-Stuff 164K, along with the tail classes performance improvements of AUCSeg compared to the runner-up method.

Table 11: The comparison of imbalance ratio and tail classes performance improvements on ADE20K, Cityscapes, and COCO-Stuff 164K.

| Dataset | ADE20K | Cityscapes | COCO-Stuff 164K |
|---|---|---|---|
| $r_m$ | 90.43 | 80.39 | 38.17 |
| Tail Classes Improvement | 1.21% | 0.75% | 0.38% |

The results suggest that the larger the imbalance degree the larger the improvements of our method. ADE20K has the largest degree imbalance, therefore gaining the most significant improvement.

## G.3 More Qualitative Results

Here we present more qualitative results on the Cityscapes, ADE20K, and COCO-Stuff 164K validation sets.

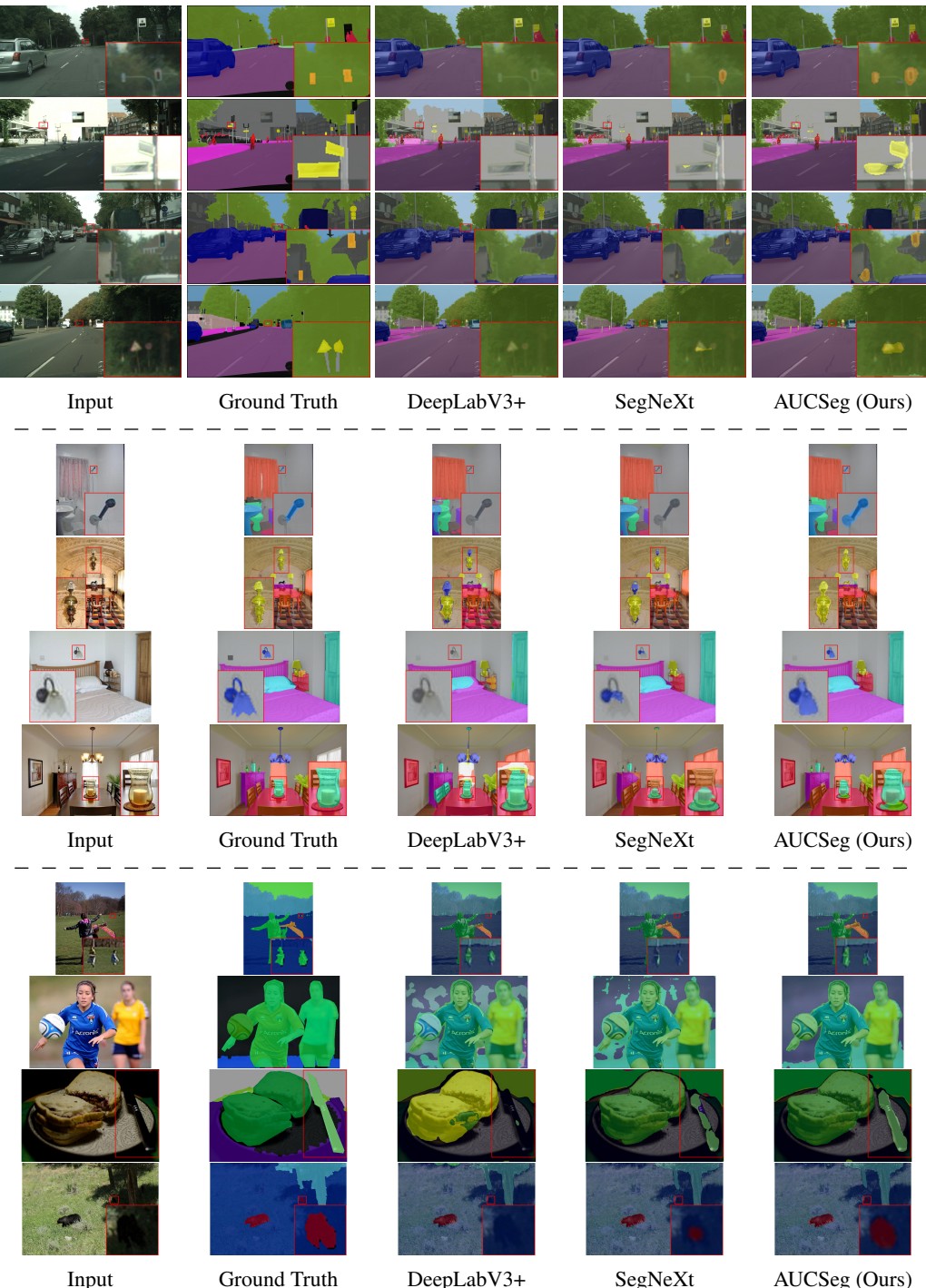

Figure 7: More qualitative results on the **Cityscapes**, **ADE20K** and **COCO-Stuff 164K** val set. Red rectangles highlight and magnify the details of the image.

### G.4 Backbone Extension of Different Model Sizes

In this paper, we use the large version of SegNeXt because of its outstanding performance. We also provide the results for the tiny, small, and base versions of SegNeXt, as shown in Table 12. The experiments indicate that AUCSeg achieves better performance under any model size.

Table 12: Results of AUCSeg on different model sizes of SegNeXt in terms of mIoU (%).

| Backbone | AUCSeg | Overall | Tail |
|---|---|---|---|
| Tiny | ✗ | 38.73 | 33.96 |
| | ✓ | **39.00** | **34.52** |
| Small | ✗ | 43.25 | 38.90 |
| | ✓ | **43.29** | **39.18** |
| Base | ✗ | 45.45 | 41.33 |
| | ✓ | **46.37** | **42.49** |
| Large | ✗ | 47.45 | 43.28 |
| | ✓ | **49.20** | **45.52** |

### G.5 Backbone Extension of Different Pixel-level Long-tail Problems

Apart from semantic segmentation, salient object detection is also a pixel-level task. In salient object detection, the salient objects often exhibit a long-tailed distribution. We apply AUCSeg to this task, using the latest SOTA method SI-SOD-EDN [81, 53] as the backbone. The results are shown in Table 13.

Table 13: Experimental results of AUCSeg in the salient object detection.

| Dataset | ECSSD | | | HKU-IS | | | PASCAL-S | | |
|---|---|---|---|---|---|---|---|---|---|
| | MAE ↓ | $F_m^\beta$ ↑ | $E_m$ ↑ | MAE ↓ | $F_m^\beta$ ↑ | $E_m$ ↑ | MAE ↓ | $F_m^\beta$ ↑ | $E_m$ ↑ |
| SI-SOD-EDN | 0.0358 | 0.9084 | 0.9375 | 0.0287 | 0.8986 | 0.9442 | 0.0644 | 0.826 | 0.8859 |
| +AUCSeg | **0.0349** | **0.9087** | **0.9377** | **0.0278** | **0.8992** | **0.9455** | **0.0629** | **0.8281** | **0.8875** |

Our AUCSeg achieves improvements across three commonly used evaluation metrics on three datasets, demonstrating that our method is highly versatile and extensible.

### G.6 Spatial Resource Consumption

In this section, we thoroughly explore the spatial resource consumption of the T-Memory Bank.

Table 14 details the ablation experiments on the spatial resource use of the T-Memory Bank. We set the memory size $S_M$ to 5 and conduct experiments on the ADE20K dataset. It is no longer necessary for samples of all classes to appear in the mini-batch, but rather only a minimum of 5 tail class samples are needed. The results demonstrate that using AUC alone requires a batch size and graphics memory 5.5 times greater than the baseline to satisfy computational demands, which is a significant expense. However, the T-Memory Bank substantially reduces this cost, enabling more efficient training without an intolerant increase in graphics memory.

Moreover, we explore the effect of memory size $S_M$ on spatial resource consumption. The findings in Figure 8 show that the graphics memory occupation increases slightly as $S_M$ rises. However, a smaller memory size generally suffices for effective performance, indicating that AUCSeg can achieve significant improvements with a manageable graphics memory burden. As shown in Figure 5a, when $S_M = 5$, significant performance improvements can be achieved with lower spatial resource consumption.

Table 14: Space resource consumption required for training properly. TMB is the abbreviation of T-Memory Bank.

| AUC | TMB | Batch Size | Graphic Memory |
|:---:|:---:|:---:|:---:|
| ✗ | ✗ | 4 | 13.29G |
| ✓ | ✗ | 22 | 72.90G |
| ✓ | ✓ | 4 | 15.45G |

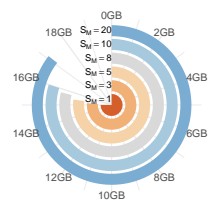

Figure 8: The effect of memory size on spatial resource consumption.

## G.7 Results of Different AUC Surrogate Losses and Calculation Methods

AUCSeg can adopt various surrogate losses. In the previous section, we use the square loss. In Table 15, we explore two other popular surrogate losses (hinge loss and exponential loss) for AUCSeg. Additionally, we include results for two AUC loss calculation methods (one-vs-one and one-vs-all) applied to AUCSeg using the square loss. The results are presented in Table 16.

Table 15: Results of different AUC surrogate losses in terms of mIoU (%).

| Dataset | AUC Surrogate Loss | Overall | Tail |
|:---:|:---:|:---:|:---:|
| ADE20K | - | 47.45 | 43.28 |
| | Hinge | 48.59(+1.14) | 44.76(+1.48) |
| | Exp | 48.86(+1.41) | 45.07(+1.79) |
| | Square | **49.2(+1.75)** | **45.52(+2.24)** |
| Cityscapes | - | 82.41 | 80.92 |
| | Hinge | 82.64(+0.23) | 81.35(+0.43) |
| | Exp | 82.45(+0.04) | 81.55(+0.63) |
| | Square | **82.71(+0.30)** | **81.67(+0.75)** |
| COCO-Stuff 164K | - | 42.42 | 40.33 |
| | Hinge | 42.52(+0.10) | 40.49(+0.16) |
| | Exp | 42.52(+0.10) | 40.53(+0.20) |
| | Square | **42.73(+0.31)** | **40.72(+0.39)** |

Table 16: Results of different AUC calculation methods in terms of mIoU (%).

| Dataset | AUC Calculation Method | Overall | Tail |
|:---:|:---:|:---:|:---:|
| ADE20K | ova | 48.46 | 44.58 |
| | ovo | **49.2** | **45.52** |
| Cityscapes | ova | 82.31 | 80.79 |
| | ovo | **82.71** | **81.67** |
| COCO-Stuff 164K | ova | 42.25 | 40.17 |
| | ovo | **42.73** | **40.72** |

The results indicate that AUCSeg shows improved performance with any of the surrogate functions. Among them, using square loss and the ovo calculation method delivers the best overall performance.

## G.8 Results of the Comparison Between PMB and TMB

There are two differences between the Pixel-level Memory Bank (PMB) and our Tail-class Memory Bank (TMB). **First**, the PMB stores pixels from all classes, whereas TMB only stores pixels from tail classes. **Second**, in TMB, the storing and retrieving processes are conducted on an entire object (we ensure that the pasted pixel forms a meaningful object). However, the PMB typically focuses on a fixed number of pixels without structural information (regardless of whether these pixels can form a complete image).

**Why do we only store tail class pixels instead of all pixels?**

Table 17 shows the average number of pixels from head and tail classes per image in the ADE20K, Cityscapes, and COCO-Stuff 164K datasets.

Table 17: The average number of pixels from head and tail classes per image.

| Dataset | ADE20K | Cityscapes | COCO-Stuff 164K |
|---------|--------|-----------|-----------------|
| Head    | 46685  | 294290    | 60157           |
| Tail    | 18977  | 31128     | 22526           |

It can be observed that the number of head class pixels in each image is 2.46 to 9.45 times greater than that of the tail classes, meaning that storing head class pixels would require significantly more memory.

Table 18 compares the performance differences between storing all and tail class pixels. It shows that the PMB, which incurs additional memory costs, performs almost the same as the TMB, and even shows a noticeable decline in the Cityscapes dataset. This is because head classes appear in almost every image (for example, in urban road datasets, it is hard to find an image without head class pixels like 'road' or 'sky'), so they do not need additional supplementation. Even if some images require supplementation of head classes, their larger pixel counts might cause them to overwrite the original tail class pixels when pasted, leading to a decline in performance. Thus, we only store tail class pixels.

Table 18: The performance differences between PMB and TMB in terms of mIoU (%).

| Dataset | ADE20K | Cityscapes | COCO-Stuff 164K |
|---------|--------|-----------|-----------------|
| PMB     | 49.09  | 82.07     | 42.66           |
| TMB     | 49.2   | 82.71     | 42.73           |

**Why it is not feasible to focus on a fixed number of pixels?**

We conduct tests on the ADE20K dataset by supplementing a fixed number of tail class pixels ($10000/20000/30000/40000$) in each image and find that compared to AUCSeg, the performance differences are $-3.00\%/-1.93\%/-0.86\%/-0.73\%$. This is because supplementing a fixed number of pixels can result in incomplete images, such as only adding the front wheel of a bicycle, therefore loss of the structural information. The model is then unable to learn complete and accurate features. Therefore, in TMB, storing and retrieving are conducted on all pixels of an entire image.

### G.9 Results of Different Memory Bank Update Strategies

In the previous section, we use the random replacement strategy to update the T-Memory Bank. We experiment with three other selection methods on the ADE20K dataset:

- **First-In-First-Out (FIFO) replacement**: Prioritizes replacing the images that were first stored in the Tail-class Memory Bank.

- **Last-In-First-Out (LIFO) replacement**: Prioritizes replacing the images that were last stored in the Tail-class Memory Bank.

- **Priority Used (PU) replacement**: Prioritizes replacing images that have previously been selected by the retrieval branch.

The results are shown in table 19.

FIFO and PU both show better performance overall and on tail classes compared to random sampling. However, LIFO, by updating only the most recently added images in the T-Memory Bank, causes the earlier images to remain unchanged. This leads to overfitting and, consequently, a decline in performance.

Table 19: Results of different memory bank update Strategies on ADE20K in terms of mIoU (%).

|  | Overall | Head | Middle | Tail |
|---|---|---|---|---|
| Random | 49.2 | **80.59** | **59.45** | 45.52 |
| FIFO | **49.35** | 80.51 | 58.71 | **45.8** |
| LIFO | 49.05 | 80.35 | 58.76 | 45.45 |
| PU | 49.21 | 80.24 | 58.73 | 45.65 |

While these complex strategies can improve performance, the gains are relatively limited. On the other hand, the random replacement method is easy to implement. Exploring more complex and effective replacement methods could be a promising direction for future work.

### G.10   Detailed Results of the Ablation Study on Hyper-Parameters

The detailed results of the ablation study on hyper-parameters are shown in Table 20, Table 21, Table 22, and Table 23.

Table 20: Ablation study on Memory Size ($S_M$) in terms of mIoU (%).

| $S_M$ | Overall | Tail |
|---|---|---|
| 1 | 48.46 | 44.68 |
| 3 | 49.09 | 45.40 |
| **5** | **49.20** | **45.52** |
| 8 | 48.81 | 45.06 |
| 10 | 48.80 | 44.99 |
| 20 | 48.63 | 44.96 |

Table 21: Ablation study on Sample Ratio ($R_S$) in terms of mIoU (%).

| $R_S$ | Overall | Tail |
|---|---|---|
| 0.01 | 48.91 | 45.13 |
| 0.04 | 48.70 | 44.97 |
| **0.05** | **49.20** | **45.52** |
| 0.07 | 48.47 | 44.75 |
| 0.1 | 48.34 | 44.53 |
| 0.15 | 47.27 | 43.30 |

Table 22: Ablation study on Resize Ratio ($R_R$) in terms of mIoU (%).

| $R_R$ | Overall | Tail |
|---|---|---|
| 0.3 | 49.07 | 45.42 |
| **0.4** | **49.20** | **45.52** |
| 0.5 | 48.89 | 45.23 |
| 0.6 | 48.01 | 44.11 |
| 0.7 | 47.47 | 43.55 |
| 1 | 44.41 | 40.26 |

Table 23: Ablation study on the weight $\lambda$ for $\ell_{auc}$ and $\ell_{ce}$ in terms of mIoU (%).

| $\lambda$ | Overall | Tail |
|---|---|---|
| 1/6 | 48.88 | 45.20 |
| 1/5 | 49.07 | 45.41 |
| 1/4 | **49.20** | **45.52** |
| 1/3 | 48.85 | 45.07 |
| 1/2 | 48.76 | 44.92 |
| 1 | 48.53 | 44.74 |

**Explanation of the reasons why performance improvement decreases as memory size increases.** This is a trade-off between diversity and learnability. When the memory size is too large, the probability of any single sample being effectively learned decreases. So the model fails to focus on important examples, and thus fails to capture their features, ultimately leading to underfitting. Conversely, if the memory size is too small, the diversity of samples is limited, which leads to overfitting. Hence, there is no free lunch for increasing the bank.

Therefore, we pursue a reasonable memory size. As shown in Table 20, we believe that a memory size of 5 is suitable in most cases. As the memory size increases/decreases, the performance slightly declines due to model overfitting/underfitting.

### G.11   Results of the Ablation Study on the Impact of Batch Size

We conduct ablation experiments on the ADE20K dataset to evaluate the impact of batch size. The results are shown in Table 24:

The performance improves as the batch size increases. Moreover, our AUCSeg is consistently effective across different batch sizes.

Table 24: Results of the ablation study on the impact of batch size in terms of mIoU (%).

| Batch Size | Overall | Tail |
|---|---|---|
| 1 | 34.73 | 29.66 |
| +AUCSeg | **40.14** | **35.74** |
| 2 | 45.5 | 41.34 |
| +AUCSeg | **46.86** | **42.93** |
| 4 | 47.45 | 43.28 |
| +AUCSeg | **49.2** | **45.52** |
| 8 | 49.35 | 45.46 |
| +AUCSeg | **49.36** | **45.53** |
| 16 | 50.07 | 46.32 |
| +AUCSeg | **50.96** | **47.03** |

## H More Discussions About AUCSeg

**Training and inference efficiency.** During training, since AUCSeg ($1.62 \pm 0.2s$ per iteration) adopts pairwise loss, it will inevitably suffer from extra complexity compared to the standard CE-based SegNeXt ($0.76 \pm 0.15s$ per iteration). Besides, during inference, since AUCSeg does not modify the model's backbone, all algorithms with the same backbone achieve similar inference efficiency ($60 \pm 5ms$ per image). Overall, AUCSeg could perform well with acceptable efficiency.

**Performance trade-off between head and tail classes.** We recognize that AUCSeg might slightly impair the performance of head classes due to an increased focus on tail classes. However, this often results in substantial improvements for tail classes, a trade-off that is generally beneficial since tail classes are typically more critical. For instance, on the Cityscapes dataset, AUCSeg exhibits a marginal decrease of $0.17\%$ in head classes but gains $0.75\%$ in tail classes compared to the runner-up, SegNeXt. Furthermore, we note that performance decreases in head classes mainly arise from misclassification at the blurry edges of objects. In contrast, gains in tail classes often stem from either the successful detection of smaller objects or the more complete detection of such objects. To summarize, on the Cityscapes datasets, detecting a new tail object like 'Traffic Lights' is far more significant than precisely detecting the edge pixels of a head class like 'sky'. Thus, we consider this trade-off highly beneficial.

## I Broader Impact

We propose a general semantic segmentation method to deal with the potential bias toward long-tail objects. For fairness-sensitive scenarios, it might be helpful to improve fairness for long-tail groups.

