# OpenReview forum: "AUCSeg: AUC-oriented Pixel-level Long-tail Semantic Segmentation"
_NeurIPS.cc/2024/Conference — NeurIPS 2024 poster_

### Official Review · Reviewer_eudJ · 2024-07-03

**Soundness:** 2
**Presentation:** 3
**Contribution:** 2
**Rating:** 5
**Confidence:** 4

**Summary:**

The authors extend AUC optimization techniques to pixel-level long-tail semantic segmentation and propose a general pixel-level AUC loss function. They decompose the loss function into inner-image and inter-image terms to decouple the interdependency, and calculate bounds to theoretically prove the effectiveness of this loss. Additionally, they design a Tail-Classes Memory Bank to reduce memory demand. Qualitative and quantitative comparisons have been made on three benchmark datasets, showing the efficiency of their method.

**Strengths:**

1. The paper is clearly organized and well written.
2. The general idea of the paper is clear, and the motivation is interesting.
3. The authors provide detailed theoretical proofs.

**Weaknesses:**

1. In the tail-class memory bank module, how do you ensure that pasted tail pixels do not completely cover any category?
2. As shown in Fig 3(b), authors argue that because images contain multiple labels, it is impossible to use stratified sampling techniques to ensure each mini-batch contains at least one sample from each class. However, I believe this view is incorrect; we can group images by category, different categories may include identical images due to multi-label nature. Then we can use stratified sampling technique for each category so that although it might sample identical images (the same sample being sampled twice as both head and tail), it still ensures each mini-batch contains at least one sample from every class,  and small batch sizes can also achieve this goal. Therefore I think there’s no need for designing a Tail-Class Memory Bank.
3. How does Tail-Class Memory Bank store data? Does it directly use ground truth or predicted results? If using ground truth directly why design Store Branch? You could directly build memory bank using ground truth. If using predictions then why not just use ground truths?
4. Pasting tail-class pixels onto original image might occlude original image classes; would this behavior affect performance?
5. This method mainly relies on SegNeXt; while sometimes environmental factors prevent exact replication of other works' results. However, SegNeXt provides training logs & pre-trained weights, why did your reproduction show significant performance gaps compared with SegNeXt's original work (e.g., nearly 4% gap on ADE20K)?
6. In Table 1, this method shows significant improvement only on ADE20K but limited gains across two other datasets suggesting poor real-world generalizability inconsistent with theoretical conclusions.
7. Currently effective methods trained over small datasets often fail scaling up large-scale models. Hence the author should further provide the results with large-scale pretrained models like SAM/CLIP etc., demonstrating broader applicability/generalization potentiality thereof.

**Questions:**

1. I am confused about Tail-Class Memory Bank design, please refer weaknesses section above explaining them accordingly.
2.  Authors should clarify certain experimental outcomes referring weakness points mentioned earlier.
3. NeurIPS Paper Checklist lacks honest responses e.g., code provision/error bars reporting/new assets do not be contained, but all marked “YES”.

**Limitations:**

The authors have addressed the limitation of this method.  This method might slightly impair the performance of head classes due to an increased focus on tail classes.

---

> ### Author Rebuttal · Authors · 2024-08-06
>
> We deeply appreciate your time and effort in providing us with such constructive comments. We would like to respond to them as follows:
>
> > **Q1:** In the tail-class memory bank module, how do you ensure that pasted tail pixels do not completely cover any category?
>
> **A1:**  The number of head class pixels in each image is 2.46 to 9.45 times greater than that of the tail classes (see the response to Reviewer `WAeT`'s `Q3`), meaning that the number of tail class pixels is too small to completely cover the head classes.
>
> We conduct 10000 random coverage experiments on the ADE20K, Cityscapes, and COCO-Stuff 164K. In the experiments, we track the number of instances where pixels of a certain category in the original image are completely covered during the random coverage process. The experimental results are as follows:
>
> | ADE20K  | Cityscapes | COCO-Stuff 164K |
> | ------- | ---------- | --------------- |
> | 7/10000 | 2/10000    | 9/10000         |
>
> The experimental results indicate that the likelihood of complete coverage occurring is less than one in a thousand.
>
> > **Q2:** An improved version of stratified sampling can ensure coverage of all classes. So why use the Tail-Class Memory Bank?
>
> **A2:** Please refer to `C-A3` in `General Response`.
>
> > **Q3:** How does Tail-Class Memory Bank store data? Does it directly use ground truth or predicted results?
>
> **A3:**  The Tail-Class Memory Bank stores a certain number of pixels for each tail class, as detailed in `Line 201-222` of the initial submission. We use ground truth to assist in storing the original image data. The storage branch is designed with three main purposes:
>
> - The storage branch uses **ground truth** to extract corresponding pixels from the original image, which is what you referred to as "directly building a memory bank using ground truth."
> - The storage branch records the original **positional information** of each pixel, which facilitates the restoration of their spatial positions by the Retrieve Branch during sampling.
> - The storage branch also assists in the **scheduling of the Memory Branch**. When the number of stored items in the Memory Branch reaches the memory size, we use a replacement method as described in the paper (see the response to Reviewer `WAeT`'s `Q4`).
>
> > **Q4:** Pasting tail-class pixels onto original image might occlude original image classes; would this behavior affect performance?
>
> **A4:** As we mentioned in `Q1`, pasting tail class pixels almost never completely covers other classes. Below, we can explore the impact of occluding the original image on performance through ablation experiments.
>
> The parameter "Resize Ratio" in the T-Memory Bank represents the scale of the image before pasting (see `Lines 220-222` in the initial submission). Ablation experiments on this parameter allow us to investigate how the area of occlusion affects performance. In the initial submission `Figure 6(c)`, we conducted this experiment on the ADE20K dataset. Below, we present additional experimental results on the Cityscapes and COCO-Stuff 164K datasets.
>
> Cityscapes:
>
> | Resize Ratio | Overall   | Tail      |
> | ------------ | --------- | --------- |
> | 0.1          | 82.18     | 80.69     |
> | 0.3          | 82.40     | 80.87     |
> | 0.5          | 82.62     | 81.50     |
> | 0.7          | **82.71** | **81.67** |
> | 0.9          | 82.53     | 81.51     |
> | 1            | 82.69     | 81.38     |
>
> COCO-Stuff 164K:
>
> | Resize Ratio | Overall   | Tail      |
> | ------------ | --------- | --------- |
> | 0.05 | 42.59     | 40.55     |
> | 0.1  | **42.73** | **40.72** |
> | 0.3 | 42.52     | 40.42     |
> | 0.5  | 41.58     | 39.79     |
> | 0.7    | 42.57     | 40.50     |
> | 1     | 42.62     | 40.64     |
>
> It can be observed that when the Resize Ratio is too large, severe occlusion occurs, resulting in a decline in model performance. Due to the differences in image sizes across datasets, the Resize Ratio should be adjusted accordingly when switching to other datasets.
>
> > **Q5:**  Why did your reproduction show significant performance gaps compared with the original work?
>
> **A5:** We used the code and parameters provided by MMSegmentation. To ensure fair comparisons, we reproduced all the experiments using a batch size of 4 instead of the original 16. The performance difference is due to the different batch sizes. We have further supplemented the experimental results with different batch sizes. Please refer to `C-A2` in the `General Response`.
>
> > **Q6:** Inconsistent performance gain.
>
> **A6:** Please refer to `C-A1` in `General Response`.
>
> > **Q7:** Comparison on large-scale pretrained models.
>
> **A7:** We extend AUCSeg to fine-tune CLIP, using DenseCLIP[1] as the backbone (batch size=4, iterations=80000). The experimental results are as follows:
>
> || Overall| Head| Middle| Tail|
> | --------------------- | --------- | --------- | --------- | --------- |
> | DenseCLIP (ResNet-50) | 32.21| 73.37| 48.49| 26.99|
> | +AUCSeg| **34.59** | **74.04** | **50.13** | **29.60** |
> | DenseCLIP (ViT-B)| 48.63| 80.36| 57.80| 45.06|
> | +AUCSeg| **49.51** | **80.70** | **59.25** | **45.91** |
>
> Our AUCSeg is also effective on large-scale pretrained models.
>
> > **Q8:** NeurIPS Paper Checklist lacks honest responses.
>
> **A8:** There may have been some misunderstandings in our responses to the Checklist questions.
>
> - In the initial submission, we provided reasons for our "yes" answers regarding code provision/new assets: "We will release the data and code once it is accepted." This might have caused some concerns. However, due to the rules during the rebuttal period, we have sent an anonymous link to the AC containing our core code (AUC Loss and T-Memory Bank), and you may need to request it from the AC.
> - When calculating the time cost for AUCSeg in the initial submission, we reported error bars. You can see this in `Lines 1178-1182` of the initial submission.
>
> ------
>
> [1] Denseclip: Language-guided dense prediction with context-aware prompting, CVPR 2022.

---

> > ### Comment · Reviewer_eudJ · 2024-08-12
> >
> > Thank you for the response. I have read your rebuttal and the comments from other reviewers, and I will maintain my original score.

---

> > > ### Author Response · Authors · 2024-08-12
> > >
> > > Thank you so much for your feedback and acceptance. Following your suggestions, we will enrich the content of our paper in the final version.

---

### Official Review · Reviewer_tNYv · 2024-07-11

**Soundness:** 2
**Presentation:** 3
**Contribution:** 2
**Rating:** 5
**Confidence:** 3

**Summary:**

This paper explores AUC optimization methods for pixel-level long-tail semantic segmentation, addressing complex dependencies and space complexity challenges. The authors propose a novel pixel-level AUC loss function and conduct a dependency-graph-based theoretical analysis to enhance generalization. They also introduce a Tail-Classes Memory Bank (T-Memory Bank) to manage memory demands, and experiments confirm the effectiveness of their AUCSeg method.

**Strengths:**

1. This paper presents a novel perspective by using AUC as the loss function.
2. The paper has performed a tail class cache to boost the performance and control the memory demand.

**Weaknesses:**

Please refer to the questions.

**Questions:**

1. For a long-tail semantic segmentation problem, which is also a pixel-wise classification task, I am uncertain why the authors chose this perspective. From the standpoint of the loss function, it emphasizes the contrast between the current class and other classes. Would a contrastive classification approach achieve similar results?
2. The significance of Proposition 1 is not very clear. What is the meaning of the $\Omega$? My understanding is that for a tail class, it is naturally difficult to randomly sample such a sample due to the scarcity. Therefore, re-sampling techniques are needed, which seems quite intuitive. What additional information is this proposition supposed to convey?
3. The logic of using a memory bank in this paper seems weired. The authors argue that it is difficult to randomly sample a tail sample, hence the need for a memory bank. Most long-tail based work considers batch re-sampling a very conventional approach. However, this paper's method enhances the diversity of tail samples, thereby improving the model's learning ability for tail samples. Perhaps rephrasing this aspect would be more effective.

**Limitations:**

Yes

---

> ### Author Rebuttal · Authors · 2024-08-06
>
> We deeply appreciate your time and effort in providing us with such constructive comments. We would like to respond to them as follows:
>
> > **Q1:** Why did the authors choose the segmentation perspective? Would a contrastive classification approach achieve similar results?
>
> **A1:** Thank you for the helpful suggestion! We will first explain why this paper focuses on the long-tailed semantic segmentation task. Then, we will discuss new findings by comparing AUC and contrastive learning from the perspective of loss functions.
>
> **Firstly**, semantic segmentation, as a pixel-level classification task, also suffers from a severe long-tail problem, yet it has been largely overlooked or insufficiently explored by the long-tail community. We approach this issue from the perspective of loss functions, aiming to find a theoretically grounded loss. Noticing that AUC is popular and effective in image-level tasks, we extend it to address the pixel-level long-tail problem.
>
> We chose to address this challenge from the loss function perspective because 1) The quality of the loss function directly impacts the model's learning quality and is directly related to performance. 2) Loss functions are more general and have the potential to be extended to other pixel-level tasks to address their long-tail issues. For example, in salient object detection, the salient objects often exhibit a long-tailed distribution. We apply AUCSeg to this task, using the latest SOTA method SI-SOD-EDN [1,2] as the backbone. See `Table 5` in the PDF submitted in the `General Response` for the results.
>
> Our AUCSeg achieves improvements across three commonly used evaluation metrics on three datasets. This demonstrates that our method is highly versatile and extensible.
>
> **Next**, we will discuss new insights comparing AUC and contrastive learning from the perspective of loss functions, supported by theoretical analysis and experimental validation.
>
> > **Theorem:** Minimizing the weighted contrastive loss approximately corresponds to minimizing an upper bound of the logistic AUC loss:
> > $$
> > \sum_{i}w_i\left[-\log\frac{e^{f(x^i)}}{e^{f(x^i)}+\sum_{j\neq i}w_je^{f(x^j)}}\right]\geq\sum_{i}\sum_{j \neq i}\frac{1}{n_in_j}\left[-\log\left(\frac{1}{1+e^{f(x^j)-f(x^i)}}\right)\right]
> > $$
> > where, $w_j=\frac{1/n_j}{\sum_{k\neq i}1/n_k}$ and $w_i=\frac{\sum_{k\neq i}1/n_k}{n_i}$.
>
> Due to space constraints, the proof will be provided in the next version. This theorem indicates that minimizing a **weighted version** of contrastive loss can implicitly optimize the OVO logistic AUC loss. The experiments in the table below also verify that Contrastive Loss + TMB and AUCSeg (Logistic) produce similar results. Our paper adopts a more general form of AUC loss, with the square surrogate loss showing the best performance.
>
> || Overall| Tail|
> |--------------------|---------|---------|
> | Contrastive Loss+TMB| 47.47| 43.55|
> | AUCSeg (Logistic)| 47.86| 43.96|
> | AUCSeg (Hinge)| 48.59| 44.76|
> | AUCSeg (Exp)| 48.86| 45.07|
> | AUCSeg (Square)|**49.20**|**45.52**|
>
> > **Q2:** The significance of Proposition 1 is not very clear, What additional information is this proposition supposed to convey?
>
> **A2:** Thank you for your question. In `Proposition 1`, $x = \Omega(y)$ means $\exists c>0, x > c \cdot y $. It represents the minimum batch size required to include pixels from all classes with a high probability. For ease of reading, we will include **a table of symbol definitions** in the final version of the paper (See `Table 1` in the PDF submitted in the `General Response`).
>
> Due to the scarcity of tail classes and the coupling between pixels, random sampling becomes challenging. This proposition quantifies the minimal batch size to cover all classes with a high probability. As we discussed in `Remark 2` of the initial submission, random sampling that meets the required conditions would necessitate sampling 759 images at once to form a mini-batch, an unbearable memory cost for model training.
>
> > **Q3:** The logic of using a memory bank in this paper seems weird. Why not use batch re-sampling? Rephrase this aspect of diversity.
>
> **A3:** Thank you for your constructive suggestion!
>
> - **The reason for not directly using batch re-sampling:** Please refer to `C-A3` in `General Response`.
> - **Why is the primary function of the T-Memory Bank not aimed at enhancing the diversity of tail samples:** As shown in `Table 2` of our initial submission (we also provide this table below), using a memory bank does indeed enhance the diversity of tail classes as an implicit way of augmentation (comparing the rows for SegNeXt and SegNeXt+TMB in the table). However, we cannot rely on the bank to fully address the long-tail issue, since the bank capacity is always limited. That's why we also need to consider the problem from the loss perspective. We find that AUC loss focuses only on the ranking loss between positive and negative samples and is not sensitive to data distribution, fundamentally avoiding the risk of underfitting caused by insufficient training samples. **We believe that the use of the TMB in our paper is intended to both facilitate the effectiveness of AUC loss and enhance tail sample diversity.**
>
> | Model | AUC  | TMB  | Overall   | Tail      |
> | ----------- | ---- | ---- | --------- | --------- |
> | SegNeXt     |   |   | 47.45| 43.28|
> | SegNeXt+AUC |yes |   | 48.46     | 44.70     |
> | SegNeXt+TMB |   | yes  | 47.86     | 43.86     |
> | AUCSeg      | yes  | yes  | **49.20** | **45.52** |
>
> We acknowledge that, to some extent, the TMB also contributes to performance improvement by enhancing tail class diversity. Therefore, in the final version of the paper, we will discuss this point further in the `Discussions` section (`Line 223` in the initial submission).
>
> ------
>
> [1] EDN: Salient object detection via extremely-downsampled network, TIP 2022.
>
> [2] Size-invariance Matters: Rethinking Metrics and Losses for Imbalanced Multi-object Salient Object Detection, ICML 2024.

---

> > ### Author Response · Authors · 2024-08-07
> >
> > We provide the proof of the theorem used in the rebuttal below:
> >
> > > **Theorem:** Minimizing the weighted contrastive loss approximately corresponds to minimizing an upper bound of the logistic AUC loss:
> > > $$
> > > \sum_{i}w_i\left[-\log\frac{e^{f(x^i)}}{e^{f(x^i)}+\sum_{j\neq i}w_je^{f(x^j)}}\right]\geq\sum_{i}\sum_{j \neq i}\frac{1}{n_in_j}\left[-\log\left(\frac{1}{1+e^{f(x^j)-f(x^i)}}\right)\right]
> > > $$
> > > where, $w_j=\frac{1/n_j}{\sum_{k\neq i}1/n_k}$ and $w_i=\frac{\sum_{k\neq i}1/n_k}{n_i}$.
> >
> > **Proof.**
> >
> > For the AUC loss under the logistic surrogate loss function:
> > $$
> > \begin{aligned}
> > \ell_{auc}^{logistic}&=\ell_{logistic}\left(f(x^+)-f(x^-)\right)\\\\
> > &=\sum_{i}\sum_{j \neq i}\frac{1}{n_in_j}\left[-\log\left(\frac{1}{1+e^{f(x^j)-f(x^i)}}\right)\right]\\\\
> > &=\sum_{i}\sum_{j \neq i}w_iw_j\left[-\log\left(\frac{1}{1+e^{f(x^j)-f(x^i)}}\right)\right]\\\\
> > &=\sum_{i}w_i\sum_{j\neq i}w_j\left[-\log(e^{f(x^i)})+\log(e^{f(x^i)}+e^{f(x^j)})\right]\\\\
> > &=\sum_{i}w_i\left[-\log(e^{f(x^i)})+\sum_{j\neq i}w_j\log(e^{f(x^i)}+e^{f(x^j)})\right]\\\\
> > &\leq \sum_{i}w_i\left[-\log(e^{f(x^i)})+\log\left(\sum_{j\neq i}w_j(e^{f(x^i)}+e^{f(x^j)})\right)\right]\\\\
> > &=\sum_{i}w_i\left[-\log(e^{f(x^i)})+\log\left(e^{f(x^i)}+\sum_{j\neq i}w_je^{f(x^j)}\right)\right]\\\\
> > &=\sum_{i}w_i\left[-\log\frac{e^{f(x^i)}}{e^{f(x^i)}+\sum_{j\neq i}w_je^{f(x^j)}}\right]
> > \end{aligned}
> > $$
> > where, $w_j=\frac{1/n_in_j}{\sum_{k\neq i}1/n_in_k}=\frac{1/n_j}{\sum_{k\neq i}1/n_k}$ and $w_i=\frac{\sum_{k\neq i}1/n_k}{n_i}$.
> >
> > This completed the proof.

---

> ### Comment · Reviewer_tNYv · 2024-08-09
> **Comments from Reviewer tNYv**
>
> Thanks for the response. I will consider slightly increasing the score because some of my concerns are being relieved.
> By the way, will the code be released after the acceptance?

---

> > ### Author Response · Authors · 2024-08-09
> >
> > Thank you so much for your feedback. Due to the rules during the rebuttal period, we have sent an anonymous link to the AC containing our core code (AUC Loss and T-Memory Bank), and you may need to request it from the AC. The full code will be released after the acceptance. If you have any further questions, we would be happy to address them.

---

### Official Review · Reviewer_TAdP · 2024-07-11

**Soundness:** 4
**Presentation:** 3
**Contribution:** 4
**Rating:** 8
**Confidence:** 5

**Summary:**

This paper investigates AUC optimization within the context of pixel-level long-tail semantic segmentation (PLSS), a complex task due to intricate loss term coupling and extensive memory requirements. Initially, the authors demonstrate the potential of AUC for PLSS from a theoretical perspective by addressing the two-layer coupling issue across loss terms. Subsequently, they propose a novel Tail-Classes Memory Bank (T-Memory Bank) to manage the significant memory demands of AUC-oriented PLSS. Finally, comprehensive experiments show the effectiveness of the proposed method.

**Strengths:**

1. This paper addresses a compelling problem with clear motivations and significant contributions, offering insightful ideas for both the AUC and segmentation communities.

2. Developing a theoretically grounded loss from an AUC perspective is novel, and the proposed T-memory bank effectively mitigates the memory burden associated with pixel-level AUC optimization.

3. The performance of the proposed AUCSeg seems promising.

**Weaknesses:**

Overall, I believe this is a qualified paper for publication after addressing the following minor concerns:

1. The notations in this paper should be carefully defined. Some key symbols are used repeatedly, such as $N$ representing both batch size (In Alg.1) and sample size (In Thm.1).

2. This paper only examines the square AUC surrogate loss without considering two other popular surrogate losses for AUC optimizations (i.e., hinge and exponential losses). It is essential to provide empirical verification by applying the ignored losses to AUCSeg.

3. In Alg.1, the batch size scale will also impact the final performance of long-tailed classifications. The authors are recommended to conduct an ablation study on this aspect.

**Questions:**

Please refer to the weakness part.

---

> ### Author Rebuttal · Authors · 2024-08-06
>
> We deeply appreciate your time and effort in providing us with such constructive comments. We would like to respond to them as follows:
>
> > **Q1:** The notations in this paper should be carefully defined. Some key symbols are used repeatedly, such as 𝑁 representing both batch size (In Alg.1) and sample size (In Thm.1).
>
> **A1:** Thank you for the suggestion! We carefully reviewed the symbol definitions in the paper. The symbol $𝑁$ representing batch size in Algorithm 1 has been changed to $N_b$. Additionally, for ease of reading, we will include **a table of symbol definitions** in the final version of the paper (See `Table 1` in the PDF submitted in the `General Response`).
>
> > **Q2:** This paper only examines the square AUC surrogate loss without considering two other popular surrogate losses for AUC optimizations (i.e., hinge and exponential losses). It is essential to provide empirical verification by applying the ignored losses to AUCSeg.
>
> **A2:** Thank you for your constructive suggestion! In this version, we have explored two other popular surrogate losses (hinge loss and exponential loss) to AUCSeg. Additionally, we have included results for two AUC loss calculation methods (one-vs-one and one-vs-all) applied to AUCSeg.
>
>
> The performance of these three surrogate losses (hinge loss, exponential loss, and square loss) is presented in the table below:
>
> | Dataset         | AUC Method | Overall          | Tail             |
> | --------------- | ---------- | ---------------- | ---------------- |
> | ADE20K          | -          | 47.45            | 43.28            |
> |                 | Hinge      | 48.59(+1.14)     | 44.76(+1.48)     |
> |                 | Exp        | 48.86(+1.41)     | 45.07(+1.79)     |
> |                 | Square     | **49.2(+1.75)**  | **45.52(+2.24)** |
> | Cityscapes      | -          | 82.41            | 80.92            |
> |                 | Hinge      | 82.64(+0.23)     | 81.35(+0.43)     |
> |                 | Exp        | 82.45(+0.04)     | 81.55(+0.63)     |
> |                 | Square     | **82.71(+0.30)** | **81.67(+0.75)** |
> | COCO-Stuff 164K | -          | 42.42            | 40.33            |
> |                 | Hinge      | 42.52(+0.10)     | 40.49(+0.16)     |
> |                 | Exp        | 42.52(+0.10)     | 40.53(+0.20)     |
> |                 | Square     | **42.73(+0.31)** | **40.72(+0.39)** |
>
> The performance of the two AUC calculation methods (ova and ovo) when using square loss is as follows:
>
> | Dataset         | AUC Method | Overall   | Tail      |
> | --------------- | ---------- | --------- | --------- |
> | ADE20K          | ova        | 48.46     | 44.58     |
> |                 | ovo        | **49.20** | **45.52** |
> | Cityscapes      | ova        | 82.31     | 80.79     |
> |                 | ovo        | **82.71** | **81.67** |
> | COCO-Stuff 164K | ova        | 42.25     | 40.17     |
> |                 | ovo        | **42.73** | **40.72** |
>
> The results indicate that AUCSeg shows improved performance with any of the surrogate functions. Among them, using square loss and the ovo calculation method delivers the best overall performance. We will include this discussion in the final version of the paper.
>
> > **Q3:** In Alg.1, the batch size scale will also impact the final performance of long-tailed classifications. The authors are recommended to conduct an ablation study on this aspect.
>
> **A3:** Please refer to `C-A2` in `General Response`.

---

> > ### Comment · Reviewer_TAdP · 2024-08-09
> >
> > The authors have addressed my concerns, I will raise my score correspondinly.

---

> > > ### Author Response · Authors · 2024-08-09
> > >
> > > We are thankful for your acceptance and constructive feedback.

---

### Official Review · Reviewer_WAeT · 2024-07-12

**Soundness:** 3
**Presentation:** 4
**Contribution:** 3
**Rating:** 5
**Confidence:** 3

**Summary:**

This paper introduces AUC optimization into the domain of long-tailed semantic segmentation. Specifically, the authors developed a pixel-level AUC loss function tailored for long-tailed semantic segmentation tasks and introduced a tail-class memory bank to address the memory demands. Additionally, the authors utilized Rademacher complexity to provide a generalization bound for AUCSeg and theoretically analyzed the potential for AUCSeg to generalize to unseen data.

The experimental evaluation in this paper assessed the performance of the proposed strategy on various datasets. Generally speaking, this paper is relatively detailed and comprehensive.

**Strengths:**

- It is novel to introduce auc optimization in long-tailed semantic segmentation and give a detailed analysis over the feasibility of this method.

- The proposed method is validated on different datasets(ADE20k, CityScapes, COCO etc.) and has been proven to be effective in tail classes.

**Weaknesses:**

There are a few concerns about this paper:

- On COCO, the performance improvement of Tail Class seems to be limited. Could the author explain the reason?

- Fig. 6(a) shows the memory size reduction of the tail-class memory bank. The results show that as the memory size increases, the performance improvement decreases. What is the reason for this?

**Questions:**

I thank the author for their detailed work. Nevertheless, there are a few questions I wonder and I hope the author could respond.

- For Tail Class Memory Bank, what would happen if we use a pixel-level memory bank? For example, prototype or similar technology, would this lead to significant performance degradation?

- For the memory bank update problem, is there a more appropriate selection method instead of random replacement?

**Limitations:**

yes

---

> ### Author Rebuttal · Authors · 2024-08-06
>
> Thanks for your constructive comments, and we would like to make the following response.
>
> > **Q1:** On COCO, the performance improvement of Tail Class seems to be limited. Could the author explain the reason?
>
> **A1:** Please refer to `C-A1` in `General Response`.
>
> > **Q2:** Fig. 6(a) shows the memory size reduction of the tail-class memory bank. The results show that as the memory size increases, the performance improvement decreases. What is the reason for this?
>
> **A2:** This is a trade-off between diversity and learnability. When the memory size is too large, the probability of any single sample being effectively learned decreases. So the model fails to focus on important examples, and thus fails to capture their features, ultimately leading to underfitting. Conversely, if the memory size is too small, the diversity of samples is limited, which leads to overfitting. Hence, there is no free lunch for increasing the bank!
>
> Therefore, we pursue a reasonable memory size. As shown in `Figure 6(a)` in the initial submission, we believe that a memory size of 5 is suitable in most cases. As the memory size increases/decreases, the performance slightly declines due to model overfitting/underfitting.
>
> > **Q3:** For Tail Class Memory Bank, what would happen if we use a pixel-level memory bank? For example, prototype or similar technology, would this lead to significant performance degradation?
>
> **A3:** Thanks so much for your suggestion! There are two differences between the Pixel-level Memory Bank (PMB) and our Tail-class Memory Bank (TMB). **First**, the PMB stores pixels from all classes, whereas our TMB only stores pixels from tail classes. **Second**, in our TMB, the storing and retrieving processes are conducted on an entire object (we ensure that the pasted pixel forms a meaningful object). However the PMB typically focuses on a fixed number of pixels without structural information (regardless of whether these pixels can form a complete image).
>
> **We will first explain why we only store tail class pixels instead of all pixels.**
>
> The table below shows the average number of pixels from head and tail classes per image in the ADE20K, Cityscapes, and COCO-Stuff 164K datasets.
>
> |Dataset|ADE20K|Cityscapes|COCO-Stuff 164K|
> |-------|------|----------|---------------|
> | Head| 46685| 294290 | 60157|
> | Tail| 18977| 31128| 22526|
>
> It can be observed that the number of head class pixels in each image is 2.46 to 9.45 times greater than that of the tail classes, meaning that storing head class pixels would require significantly more memory.
>
> **The table below** compares the performance differences between storing all and tail class pixels. It shows that the PMB, which incurs additional memory costs, performs almost the same as the TMB, and even shows a noticeable decline in the Cityscapes dataset. This is because head classes appear in almost every image (for example, in urban road datasets, it is hard to find an image without head class pixels like 'road' or 'sky'), so they do not need additional supplementation. Even if some images require supplementation of head classes, their larger pixel counts might cause them to overwrite the original tail class pixels when pasted, leading to a decline in performance. Thus, we only store tail class pixels.
>
> | Dataset | ADE20K    | Cityscapes | COCO-Stuff 164K |
> | ------- | --------- | ---------- | --------------- |
> | PMB     | 49.09     | 82.07      | 42.66           |
> | TMB     | **49.20** | **82.71**  | **42.73**       |
>
> **Next, we will explain why it is not feasible to focus on a fixed number of pixels.**
>
> We conduct tests on the ADE20K dataset by supplementing a fixed number of tail class pixels (10000/20000/30000/40000) in each image and find that compared to AUCSeg, the performance differences are -3.00%/-1.93%/-0.86%/-0.73%. This is because supplementing a fixed number of pixels can result in incomplete images, such as only adding the front wheel of a bicycle, therefore loss of the structural information. The model is then unable to learn complete and accurate features. Therefore, in our TMB, storing and retrieving are conducted on all pixels of an entire image.
>
> We will include this part in the final version of the paper.
>
> > **Q4:** For the memory bank update problem, is there a more appropriate selection method instead of random replacement?
>
> **A4:** Thank you for your constructive suggestion! Based on your suggestion, we have now tried three other different selection methods on the ADE20K dataset:
>
> - **First-In-First-Out (FIFO) replacement**: Prioritizes replacing the images that are first stored in the Tail-class Memory Bank.
> - **Last-In-First-Out (LIFO) replacement**: Prioritizes replacing the images that are last stored in the Tail-class Memory Bank.
> - **Priority Used (PU) replacement**: Prioritizes replacing images that have previously been selected by the retrieve branch.
>
> The experimental results are shown in the table below.
>
> |        | Overall   | Head      | Middle    | Tail      |
> | ------ | --------- | --------- | --------- | --------- |
> | Random | 49.20     | **80.59** | **59.45** | 45.52     |
> | FIFO   | **49.35** | 80.51     | 58.71     | **45.80** |
> | LIFO   | 49.05     | 80.35     | 58.76     | 45.45     |
> | PU     | 49.21     | 80.24     | 58.73     | 45.65     |
>
> FIFO and PU both show better performance overall and on tail classes compared to random sampling. However, LIFO, by updating only the most recently added images in the T-Memory Bank, causes the earlier images to remain unchanged. This leads to overfitting and, consequently, a decline in performance.
>
> While these complex strategies can improve performance, the gains are relatively limited. The random replacement method, on the other hand, is easy to implement. We will include this part in the final version of the paper. In the future, we will explore more complex and effective replacement methods, hoping to provide a direction for other researchers to explore as well.

---

> > ### Comment · Reviewer_WAeT · 2024-08-13
> >
> > I appreciate the author's detailed response. Some of my concerns have been addressed. I am going to maintain my original score.

---

> > > ### Author Response · Authors · 2024-08-13
> > >
> > > Thank you so much for your feedback and acceptance. Following your suggestions, we will enrich the content of our paper in the final version. If you have any further questions, we would be happy to address them.

---

### Author Rebuttal · Authors · 2024-08-06

**General Response**

Dear SAC, AC, and reviewers,

Thank you for your invaluable feedback. Based on your comments, we have revised the details and now offer a summary of our responses.

- **Additional Experiments:**
  1. Different sampling methods for the Tail-class Memory Bank
  2. Different AUC surrogate losses and AUC calculation methods
  3. New ablation experiments for the hyperparameters
  4. Extend AUCSeg to other pixel-level long-tail tasks
- **Explanations of the Method and Theory:**
  1. The motivation for introducing the Tail-class Memory Bank, its mechanism, and why it is effective
  2. Theoretical comparison between contrastive and AUC loss
  3. Detailed analysis of dataset distribution to further explain the variations in experimental results across different datasets
- **Organization:**
  1. Reorganize the symbol definitions and include a table of symbol definitions
  2. Update our responses to the checklist

Below, we provide responses to some **Common Questions**:

------

> **C-Q1:** [Reviewer `WAeT` and `eudJ`] The performance improvement of the tail class varies across the three datasets. ADE20K shows a significant improvement, while the other two show relatively limited results. Could the author explain the reason?

**C-A1:** The performance gain depends on the degree of imbalance of the underlying dataset. To see this, we show the pairwise mean imbalance ratio $r_{m}$ (average the imbalance ratio of each class pair).

In the table below, we compare $r_{m}$ for ADE20K, Cityscapes, and COCO-Stuff 164K, along with the tail classes performance improvements of AUCSeg compared to the runner-up method.

| Dataset                  | ADE20K | Cityscapes | COCO-Stuff 164K |
| ------------------------ | ------ | ---------- | --------------- |
| $r_{m}$                  | 90.43  | 80.39      | 38.17           |
| Tail Classes Improvement | 1.21%  | 0.75%      | 0.38%           |

The results suggest that the larger the imbalance degree the larger the improvements of our method. ADE20K has the largest degree imbalance, therefore gaining the most significant improvement.


> **C-Q2:** [Reviewer `TAdP` and `eudJ`] The author should include ablation experiments for the batch size.

**C-A2:** We conduct ablation experiments on the ADE20K dataset to evaluate the impact of batch size. The results are shown below:

| Batch Size | Overall   | Tail      |
| ---------- | --------- | --------- |
| 1          | 34.73     | 29.66     |
| +AUCSeg    | **40.14** | **35.74** |
| 2          | 45.50     | 41.34     |
| +AUCSeg    | **46.86** | **42.93** |
| 4          | 47.45     | 43.28     |
| +AUCSeg    | **49.20** | **45.52** |
| 8          | 49.35     | 45.46     |
| +AUCSeg    | **49.36** | **45.53** |
| 16         | 50.07     | 46.32     |
| +AUCSeg    | **50.96** | **47.03** |

The performance improves as the batch size increases. Moreover, our AUCSeg is consistently effective across different batch sizes. In the initial submission, we selected a batch size of 4. Although this does not yield the highest performance, this batch size ensures that our experiments can be replicated on any GPU with 24GB of memory (such as the NVIDIA 3090 or 4090). We will include this part in the final version of the paper.

> **C-Q3:** [Reviewer `tNYv` and `eudJ`] Why design a Tail-class Memory Bank instead of using an improved version of stratified sampling for batch re-sampling?

**C-A3:** Stratified sampling can hardly cover all the involved classes with a small batch size (such as 4 in our paper). To ensure coverage, one has to employ a much larger batch size, producing a much higher computational burden. **Even if we can cover the classes**, tail class images may appear repeatedly, leading to overfitting and therefore a performance degradation. Our Tail-class Memory Bank, however, only involves pasting a portion of one image onto another, which behaves like an implicit data augmentation. It mitigates the sampling problem without sacrificing the generalization ability. The following empirical results demonstrate our belief.

**First**, we counted the number of images containing pixels from each class in the Cityscapes, ADE20K, and COCO-Stuff 164K datasets. See `Table 2-Table 4` in the PDF submitted in the `General Response`.

The results show that images with tail class pixels are very limited. Particularly in the ADE20K, there are only 41 images in the training set of 20210 images that contain pixels with the tail class ID of 97. The tail class images will be repeatedly sampled when using stratified sampling, leading to overfitting to such repeated images.

**Next**, we train on the ADE20K dataset following the stratified sampling method. The experimental results are as follows:

|                     | Overall   | Head      | Middle    | Tail      |
| ------------------- | --------- | --------- | --------- | --------- |
| Stratified Sampling | 46.20     | 79.55     | 57.65     | 42.21     |
| T-Memory Bank       | **49.20** | **80.59** | **59.45** | **45.52** |

The overfitting of stratified sampling brings a performance reduction for both head and middle classes. The tail classes experience an even more significant performance drop due to the heavy sample repetition in the batch. However, our T-Memory Bank, with its random pasting approach, diversifies the background of the tail classes, thus helping the model to better learn the features of the tail classes.

------

Please refer to the specific responses below for more information. We will update all these improvements in the next version.

---

### Decision · Program_Chairs · 2024-09-25

**Decision:**

Accept (poster)

**Comment:**

This paper proposes an AUC optimization method for long-tail semantic segmentation. A pixel-level AUC loss function is proposed along with a theoretical analysis, and the Tail-Class Memory Bank strategy to manage memory issues. Most of the reviewers’ comments were addressed in the rebuttal and all the reviewers were on the positive side after rebuttal. The AC agrees with the consensus among the reviewers, and recommends to accept it.